# Time-calibrated phylogenetic and chromosomal mobilome analyses of *Staphylococcus aureus* CC398 reveal geographical and host-related evolution

An international collection of *Staphylococcus aureus* of clonal complex (CC) 398 from diverse hosts spanning all continents and a 30 year-period is studied based on whole-genome sequencing (WGS) data. The collection consists of publicly available genomic data from 2994 strains and 134 recently sequenced Swiss methicillin-resistant *S. aureus* (MRSA) CC398 strains. A time-calibrated phylogeny reveals the presence of distinct phylogroups present in Asia, North and South America and Europe. European MRSA diverged from methicillin-susceptible *S. aureus* (MSSA) at the beginning of the 1950s. Two major European phylogroups (EP4 and EP5), which diverged approximately 1974, are the main drivers of MRSA CC398 spread in Europe. Within EP5, an emergent MRSA lineage spreading among the European horse population (EP5-Leq) diverged approximately 1996 from the pig lineage (EP5-Lpg), and also contains human-related strains. EP5-Leq is characterized by staphylococcal cassette chromosome *mec* (SCC*mec*) IVa and *spa* type t011 (CC398-IVa-t011), and EP5-Lpg by CC398-SCC*mec*Vc-t011. The lineage-specific antibiotic resistance and virulence gene patterns are mostly mediated by the acquisition of mobile genetic elements like SCC*mec*, *S. aureus* Genomic Islands (SaGIs), prophages and transposons. Different combinations of virulence factors are present on *S. aureus* pathogenicity islands (SaPIs), and novel antimicrobial resistance gene containing elements are associated with certain lineages expanding in Europe. This WGS-based analysis reveals the actual evolutionary trajectory and epidemiological trend of the international MRSA CC398 population considering host, temporal, geographical and molecular factors. It provides a baseline for global WGS-based One-Health studies of adaptive evolution of MRSA CC398 as well as for local outbreak investigations.

*Staphylococcus (S.) aureus* is a Gram-positive coccoid bacterium commonly colonizing the skin and mucous membranes of humans and several animal species[1,2]. It is also recognized as one of the most important opportunistic bacterial pathogens in humans and causes a wide variety of infections, such as food-borne intoxication, skin and soft tissue infections, and life-threatening bacteremia[3,4]. Animals and humans are frequently asymptomatic carriers of *S. aureus* but can also develop a wide range of different infections and may act as a turntable

✉ e-mail: vincent.perreten@unibe.ch

for human transmission[1,5]. The pathogenicity of *S. aureus* is due to the production of a variety of virulence factors, such as hemolysins, leukocidins, exfoliative toxins, proteases, super-antigenic enterotoxins and toxic-shock syndrome toxin-1 (TSST-1), some of which are commonly carried by mobile genetic elements (MGEs), such as prophages or genomic islands (SaGIs), acquired via horizontal gene transfer. The immune evasion cluster (IEC) is a set of human-specific immune-modulating proteins, such as staphylokinase (SAK), enterotoxin (SEA), chemotaxis-inhibitory protein (CHP), and staphylococcal complement inhibitor (SCN), which are commonly encoded by *hlb*-converting lysogenic phages of the *Siphoviridae* family and mediate adaptation to the human host[3,6]. SaGIs can be integrated at different locations within the genome of *S. aureus* and might carry accessory genes, especially those encoding antibiotic resistance (*S. aureus* Resistance Islands [SaRIs]) or virulence proteins (*S. aureus* Pathogenicity Islands [SaPIs]). SaPI-encoded variants of the von Willebrand factor-binding protein (vWBP) play an important role in host adaptation[1,7,8]. Furthermore, *S. aureus* can acquire antibiotic-resistance genes, posing therapeutic challenges in healthcare institutions and veterinary settings[1,4]. Of major concern is methicillin-resistant *S. aureus* (MRSA), which is resistant to all beta-lactam antibiotics except for 5th-generation cephalosporins and is frequently resistant to other clinically important antimicrobial agents[9–11]. MRSA is characterized by the acquisition of the methicillin resistance genes *mecA* or *mecC*, which are encoded by Staphylococcal Cassette Chromosome *mec* (SSC*mec*) elements[12–14]. In animals, MRSA was initially acquired from humans; however, in the 2000s, a new lineage of the MRSA clonal complex (CC) (398) emerged through adaptation in livestock and is now known as livestock-associated MRSA (LA-MRSA)[1,15]. LA-MRSA animal adaptation was characterized by the occurrence of two recombination events, namely, the acquisition of the tetracycline resistance gene *tet*(M) and the loss of the prophage-mediated IEC[16]. However, the reacquisition of IEC determinants mediated by prophages or SaPIs is frequent, allowing for human readaptation[17]. In recent decades, the prevalence of the MRSA sequence type (ST)398 in livestock has increased steadily worldwide[18–22]. Pigs and, to a lesser extent, cattle are usually asymptomatic carriers of LA-MRSA ST398 and are considered the main reservoirs[15]. Horses, on the other hand, are particularly predisposed to develop postsurgical MRSA ST398 infections[23]. Animal owners and people working with animals are at risk of being colonized, creating a new biological environment of human-adapted clones that are currently also associated with clinical infections[5,24–30].

In this study, the reconstruction of a time-calibrated phylogeny in combination with geographical and epidemiological data from more than 3000 strains was used to determine when and where successful lineages may have emerged. Additionally, an in silico screening for the presence of virulence and antimicrobial resistance genes was used to determine whether specific patterns were specifically associated with certain lineages and hosts. Representative strains of the major European lineages from the Swiss collection were hybrid assembled, and an in-depth analysis of the chromosome-associated mobilome, i.e., prophages and genomic islands, was performed to identify mobile genetic elements causing the gene carriage patterns observed. This approach, based on a deeper analysis of the epidemiological spread and recombination events involved in the acquisition of resistance and virulence genes, provides new insights into the temporal emergence and evolution of different lineages of MRSA CC398 at a global scale and will serve as an updated baseline for future analysis.

## Results

### CC398 population structure: time-calibrated phylogeny, molecular characterization, host, and geographic distribution

Based on our analysis, the molecular clock of the CC398 lineage was estimated to be $3.34 \times 10^{-7}$ substitutions per site per year [95% higher posterior density (HPD) interval: $3.20–3.48 \times 10^{-7}$] which is consistent with but lower than that estimated in previous studies for *S. aureus*[31–33] and specifically CC398[34]. This phylogenetic analysis revealed the presence of six clades, seven major phylogroups, and several lineages and sublineages (Fig. 1).

Clade one (c1) ($n = 64$) was formed mostly by Asian MSSA (only five MRSA) and was the most distant from the current MRSA clades. The divergence time from the most recent common ancestor of the remaining clades was estimated to be approximately 1949 [95% C.I.: 1942–1956]. The strains were isolated from different host species and sample materials, the majority carried AMR genes against several antibiotic classes and IECs were carried by less than 25% of the strains in this clade (most common *chp-scn*) (Fig. 2 and Supplementary Data 1).

Clade two (c2) ($n = 591$) was also formed by MSSA distant from the MRSA clades, from which they diverged around the beginning of the 1950s [95% C.I.: 1947–1960]. It is composed of a major phylogroup of MSSA commonly isolated from human carriers in Europe and North America (called the susceptible phylogroup–SP), which are predominantly *spa* type t571 (t1451 predominant in Mexico), of which the majority (92%) carry the MLS$_B$ resistance gene *erm*(T) and an IEC (*chp-scn* combinations) (95%) which are traits indicative of the human-adapted clade.

Clade three (c3) ($n = 355$) evolved initially in Asia, where it later split approximately 1957 [95% C.I.: 1952–1964], giving rise to two distinct Asian subclades. The first one comprises a phylogroup of MRSA isolated mostly from human clinical samples in China ($n = 148$, Asian phylogroup–AP), which were predominantly *spa* type t034 ($n = 107$) and carried a *chp-sak-scn*-encoding IEC. The second subclade was instead formed by MSSA isolated from human carriers and animals (cattle and pigs); this subclade showed no *spa* type specificity and carried an IEC (*chp-scn*). Crucially, the first European MRSA phylogroup ($n = 77$, EP1), which was mostly isolated from human clinical samples in Denmark (also three from the Netherlands, two from Hungary, and four from New Zealand), evolved from the MSSA subclade and, specifically, from a susceptible strain isolated in the UK approximately 1972 [95% C.I.: 1965-1980]. Notably, all the strains from EP1 harbored an IEC (*chp-sak-scn*), the PVL, and lacked the *tet*(M)-carrying transposon Tn*916*, which are characteristic features of the so-called CC398 human-adapted clade[17]. The EP1 strains also carried the resistance genes *mecA* and *blaZ*, *ant(9)-Ia*, *erm*(A), and *tet*(K), which confer resistance to beta-lactams, aminoglycosides, macrolides-lincosamides-streptogramin B (MLS$_B$), and tetracyclines, respectively. A lineage from EP1 containing 39 strains from Denmark that emerged in approximately 2010 also acquired the trimethoprim resistance gene *dfrG* (Fig. 2).

Between 1957 and 1956 approximately, the jump from humans to livestock which is characterized by the acquisition of the *tet*(M) gene and the loss of the φSa3 *hlb*-converting prophage encoding the IEC took place, and subsequently all the LA-MRSA CC398 clades and phylogroups identified in the study emerged[35]. Clade five (c5) ($n = 159$) emerged around 1965 [95% C.I.: 1961–1969] and consisted of the denominated European phylogroup 2 (EP2), with mostly MRSA strains (predominantly of *spa* type t108) that have acquired an SCC*mec* of type V. These strains were detected among human carriers in the Netherlands; however, they did not seem to have undergone human adaptation since none of them harbored an IEC.

Around 1974 [95% CI: 1971–1978] the remaining clade (c6), which is the main driver of CC398 spread in Europe, diverged forming three branches which included the remaining European MRSA phylogroups, EP3, EP4, and EP5 (Figs. 1 and 2).

The first branch of clade 6 further split forming two phylogroups, the AAP (American-Asian phylogroup) and EP3. AAP ($n = 61$) consisted of MSSA and MRSA (predominantly *spa* type t571) isolated mostly from pigs in Asia and North America which did not contain an IEC or a PVL but rather a plethora of resistance genes (Supplementary Data 1). EP3

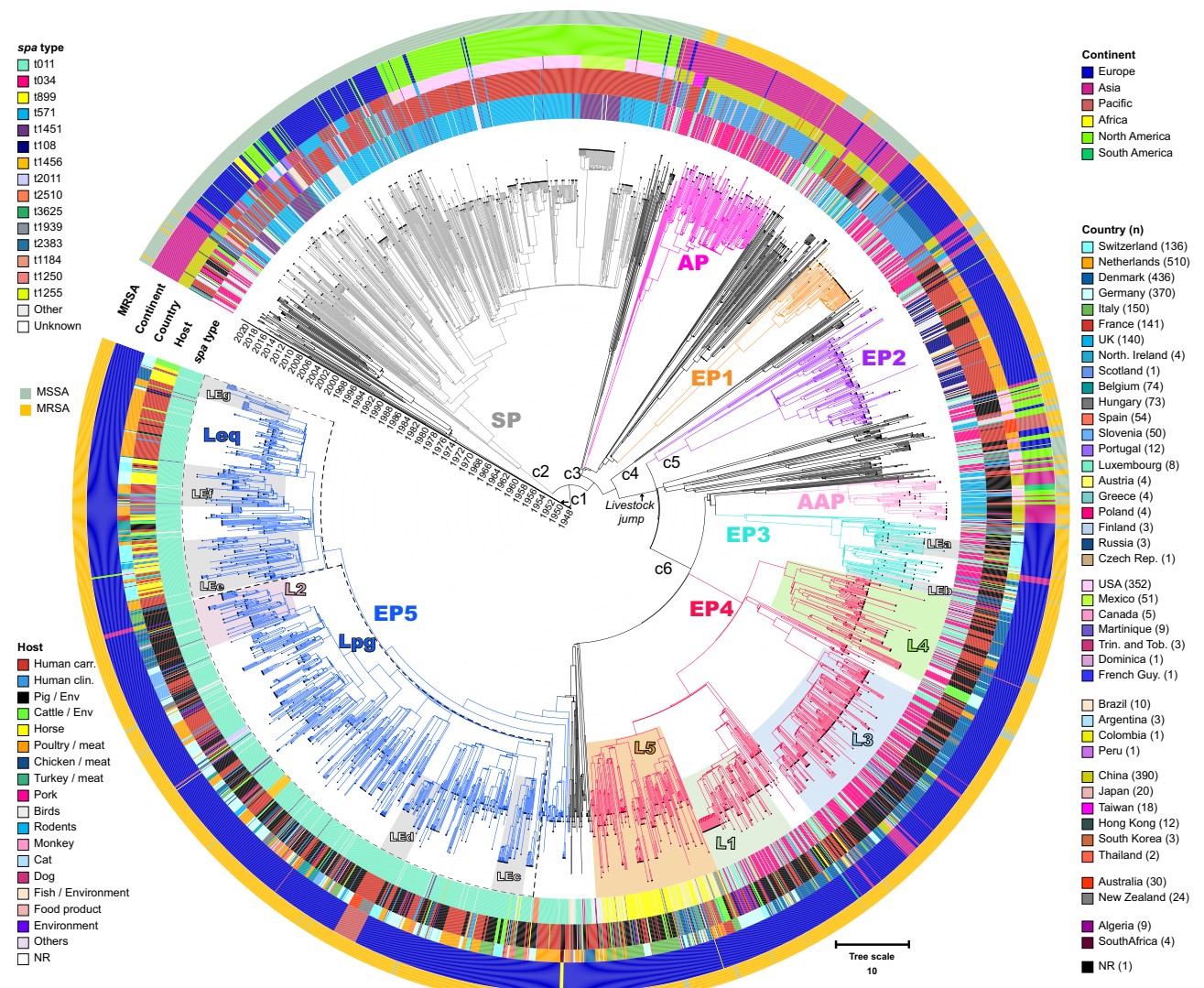

**Fig. 1 | Time-calibrated phylogeny of an intercontinental collection of 3128 MSSA and MRSA CC398 isolates.** The 3128 isolates were compared pairwise to the MRSA ST398 type strain reference genome (GenBank accession number NC_017333.1) and a cgSNP alignment was constructed after calling SNPs and removing recombination events, this alignment was used in combination with the isolation date to perform a MCMC simulation and reconstruct a Bayesian inferred phylogeny. Colored branches correspond to phylogroups; namely gray SP, dark pink AP, light pink AAP, orange EP1, violet EP2, turquoise EP3, red EP4, and blue EP5. carr. carriage, clin. clinical, c1-6: corresponds to major clades 1 to 6. Lpg: pig-associated lineage. Leq: equine-associated lineage. L1-5: major European lineages based on previous studies. LEa-g: minor lineages. External rings (from outer to inner one) represent metadata from the samples, namely MRSA, continent, country of isolation, host species, and *spa* type, respectively.

($n = 104$) contained strains belonging predominantly to either *spa* type t034 or t011 that were isolated mostly from pigs, cattle, chickens, and humans in several European countries ($n = 97$), such as Switzerland, Germany, Denmark, the Netherlands, Hungary, and Italy but also in China ($n = 7$). These strains harbored a specific pattern of resistance genes consisting of *blaPC1, ant(9)-Ia, erm*(A)*, vga*(E)*, tet*(M)*, tet*(K) and *dfrG*, and only six strains harbored IECs with four different gene combinations (one with only *chp*, one with *sak-scn-selp*, one with *chp-sak-scn*, and three with *sak-scn*), suggesting individual and isolated reacquisition events of IEC-carrying prophages.

The second branch of c6 corresponded to the European phylogroup 4 (EP4; $n = 652$) that further diverged into four lineages and several sublineages, including the so-called Danish (also isolated in other countries) pig-associated lineages L1, L3, and L4 and the Italian lineage of *spa* t899 (L5) (names of lineages with geographical association were previously defined as such)[36]. The four lineages of EP4 presented diverse profiles of AMR and virulence. L1 strains predominantly displayed a conserved pattern of genes, including *aadD*,

*ant(6)-Ia, spw, lnu*(B)*, lsa*(E)*, tet*(M)*, tet*(K), and *dfrG*, as well as mutations in the fluoroquinolone resistance-determining region of *gyrA* and *grlA*. Interestingly, a sublineage of L1 evolving in Germany additionally harbored *erm*(C)*, dfrK*, and a *tarP*-carrying Sa1int prophage (Fig. 2). The L3 strains exhibited a conserved resistance pattern with *blaPC1, ant(9)-Ia, lnu*(B)*, lsa*(E)*, tet*(M)*, tet*(K) and *dfrG*. Lineage L4 strains carried several different AMR genes and no specific conserved pattern. L5 was composed mostly of *spa* type t899 strains, which predominantly carried the *blaPC, vga*(A)*, and *tet*(M) resistance genes. It evolved into two sublineages that spread among different host species through Europe: one sublineage characterized by strains carrying a SCC*mec* type IV and fluoroquinolone resistance-associated mutations in *grlA* and by the presence of the IEC (*chp-sak-scn*) in 84% ($n = 54$) of them (Fig. 2). The second sublineage evolved locally in Germany, the Netherlands, and Italy; contained strains without IECs; and harbored either SCC*mec* type IV or SCC*mec* type V (Fig. 2).

From the third branch of c6 emerged around 1984 the larger European phylogroup (EP5, $n = 1000$) which included two major

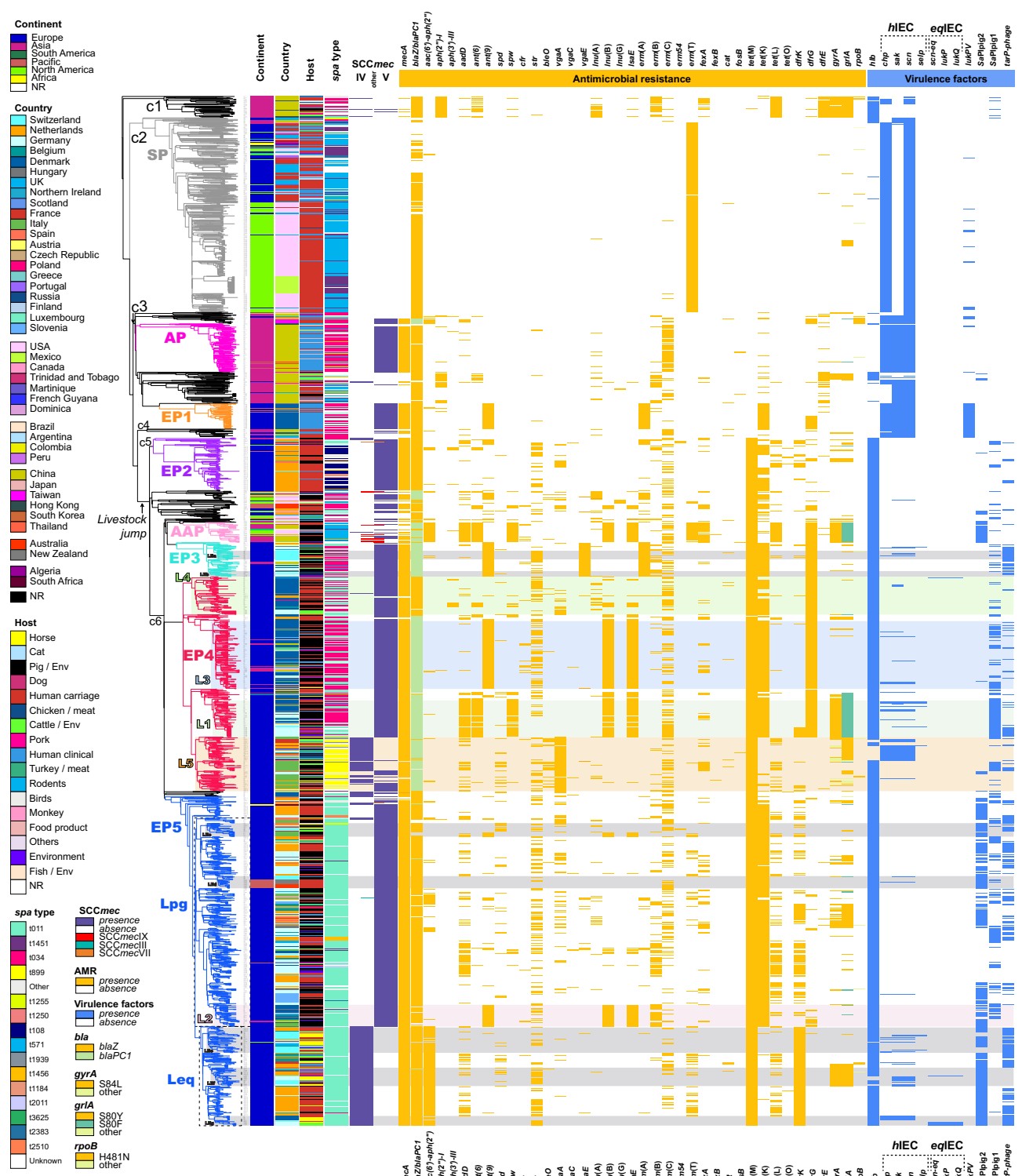

**Fig. 2 | Time-calibrated phylogenetic distribution, molecular characterization, antimicrobial resistance, and partial virulence profile of 3128 *S. aureus* strains of CC398.** In silico screening of AMR, virulence (Seemann T, Abricate, https://github.com/tseemann/abricate), *spa*[87], and SCC*mec* typing[71]. SCC*mec mec*-containing staphylococcal chromosomal cassete, *h*IEC human-associated immune

evasion cluster, *eq*IEC equine-associated immune evasion cluster, SaPI *S. aureus* pathogenicity islands, c1-6 clades 1 to 6, SP susceptible phylogroup, AP Asian phylogroup, AAP American-Asian phylogroup, EP1-5 European phylogroups 1 to 5, Lpg pig-associated lineage, Leq equine-associated lineage, L1-5 major European lineages based on previous studies, LEa-g minor lineages.

lineages, each composed of several sublineages. EP5 was characterized by the predominance of *spa* type t011 strains and, in several sublineages, the presence of SaPIPig2 (*n* = 549), which carries the *scn* and *vwb* genes. The two EP5 lineages diverged approximately 1993 [95% C.I.: 1991–1995]. The first one contained strains isolated from pigs

(*n* = 334) but also from human carriers (*n* = 250) and was defined as a pig-associated lineage (Lpg, *n* = 634). Most of the strains (>90%) acquired a SCC*mec* of type V and contained *blaZ*, *tet*(M), and *tet*(K). The majority of sublineages belonging to Lpg contained strains that were identified in several European countries, such as Denmark,

Finland, Switzerland, Germany, the Netherlands, Italy, Hungary, Belgium, the UK, France, Austria, Spain, Slovenia, Greece, Poland, and Luxembourg. However, one sublineage (LEd) of Lpg related to a Swiss sublineage was formed by strains from non-European territories, including strains from New Zealand ($n = 15$) and Australia ($n = 11$); half of the strains of this sublineage carried the *erm*(C) gene. One of the sublineages of Lpg contained strains with additional resistance genes (*ant(9)-Ia*, *lnu*(B), *lsa*(E), and *erm*(B)) and consisted of the already defined Danish L2 ($n = 66$, mostly associated with pigs). Our analysis revealed that L2 contained not only strains from Denmark ($n = 30$, from pigs, humans and cattle) but also from other European countries, such as Italy ($n = 13$, pig-associated), Hungary ($n = 9$, environment and humans), Belgium ($n = 4$, pigs), Finland ($n = 2$, pigs), Germany ($n = 2$, pigs), Slovenia ($n = 1$, human), and five strains from pigs in China.

The second major lineage of EP5 was named Leq (equine lineage, $n = 301$) since horses represented the principal animal species carrying strains ($n = 75$) of this lineage. A considerable number of human strains mostly originated from people working with horses as well as from human clinical samples (carriage, $n = 137$; clinical, $n = 30$). Leq strains were identified in the Netherlands ($n = 130$, 95% of which were isolated from humans); Switzerland ($n = 59$, with 29 from humans and 24 from horses); Germany ($n = 39$, where it was also reported in cattle); Belgium ($n = 29$, 10 from pigs, six from cattle, five from humans and chickens); France ($n = 14$ from humans, horses and dogs); Denmark ($n = 13$ from horses and human carriers); the UK ($n = 11$ from horses, humans and cattle); Spain (n = 1 from horse); Italy ($n = 1$ from pig); and the USA ($n = 2$ from horse and donkey). Leq strains were characterized by a SCC*mec* type IV and a conserved pattern of resistance genes (*blaZ*, *aac(6')-Ie-aph(2'')-Ia*, *tet*(M), and *dfrK*), as well as by the presence of SaPIpig2 in the majority ($n = 266$) of them and a Sa1int-*tarP* carrying prophage in 171 strains (see Comparative Genomics). Notably, the emergence of Leq sublineages with strains from different countries harboring mutations in the fluoroquinolone resistance-determining region of the *gyrA* and *grlA* genes ($n = 66$) and strains in Denmark that lost SaPIpig2 (11 out of 13) and harbored a *chp-sak-scn* IEC, as well as strains from horses and cattle in Germany that also carried the prophage φSaeq1-associated elements *scn-eq*, *lukP* and *lukQ*, shows the ongoing evolution and adaptation of this lineage to new niches (Fig. 2).

A closer analysis of the Swiss MRSA strains within the international phylogeny revealed that all but two ($n = 133$) of the strains belonged to the LA-MRSA CC398 clade 6, with the majority clustering within phylogroups EP3 and EP5. Together with other international strains, they formed sublineages LEc, LEd, LEe, LEf, and LEg within EP5 and LEa and LEb within EP3. LEe, LEf, and LEg belong to the denominated equine lineage (Leq) of EP5. LEe and LEf split approximately 1998 [95% C.I.: 1997-2000]. LEe ($n = 77$) contained strains isolated from humans, cattle, and pigs from several countries, such as Switzerland ($n = 46$, 22 from horses, 21 from humans), the Netherlands ($n = 16$, 14 from humans, and two from rodents), the UK ($n = 6$, horses and human clinical samples), France ($n = 3$, dog and horses), Belgium ($n = 3$, horse, pig, and human clinical sample), the USA ($n = 2$, horse and donkey) and Germany ($n = 1$, cattle). LEf ($n = 56$) included Swiss ($n = 12$, seven from humans, two from horses, one from a pig, one from a dog, and one from the environment) and German ($n = 21$, 20 isolated from horses and one from turkey) strains. LEg ($n = 29$) was formed by strains from Denmark ($n = 13$, 10 from horses and 3 from human carriers), Germany ($n = 13$, 5 from horses, 7 from cattle, and 1 from a human carrier), and single isolates from Spain (horse), the UK (human clinical) and Switzerland (human carrier). LEc and LEd clustered within the pig lineage (Lpg) of EP5 and contained strains from other animals (e.g., cattle, horses, chickens, rodents) and humans. The LEc split approximately 2002 [95% C.I.: 2001-2004] into three branches, the first consisting of strains from France ($n = 2$) (human carriage and pig), the second consisting of only strains from Switzerland ($n = 12$) (9 from cattle and 3

from pigs) and the third consisting of strains from Italy ($n = 15$, pigs) and France ($n = 1$, cattle). The sublineage LEd was formed by two branches that diverged approximately 2007 [95% C.I.: 2005-2010]. The first branch of LEd consisted of strains isolated from pigs ($n = 7$) and humans ($n = 4$) in Switzerland, and the second branch consisted of strains from human carriers in New Zealand ($n = 15$) and Australia ($n = 11$) (Fig. 2).

The two lineages of EP3 (LEa and LEb) have different epidemiological backgrounds. LEa contained only strains isolated in Switzerland from pigs ($n = 10$), cattle ($n = 6$), humans ($n = 7$) and pork meat ($n = 2$), and LEb contained strains from diverse animals, with half of them isolated from chickens ($n = 8$) in Switzerland, Germany, and the Netherlands. Less predominant Swiss strains (which may indicate isolated introduction events) belonged to sublineages L1 ($n = 4$, 3 from cattle and 1 from pig), L3 ($n = 3$, 2 from chicken and 1 from human), L5-t899 ($n = 1$, human carriage) and LEg ($n = 1$, human carriage); this lineage had one branch isolated in Denmark and one predominantly in Germany. Two clinical MRSA strains from humans in Switzerland clustered within the distant phylogroups EP2 and AP, representing isolated events.

## Pangenome-based population structure

The reconstructed phylogeny and its respective estimated parameters (i.e., molecular clock and population model) already showed that the CC398 *S. aureus* population structure, especially that of MRSA, which is widely distributed throughout Europe, is very homogeneous, with an estimated low mean mutation rate of $3.3422 \times 10^{-7}$ (95% C.I.: $3.0783 \times 10^{-7}$–$3.6188 \times 10^{-7}$) substitutions per site per year and few nucleotide differences between phylogroups and lineages. Pangenome analysis of the entire collection revealed a total of 6317 genes (Fig. 3a), and the core genome (≥95%) consisted of 2246 loci, representing 385 loci more than the cgMLST scheme publicly available for *S. aureus* (1861 loci). The accessory genome consisted of 3669 shell genes (present in two or more strains) and 402 cloud genes that were present in only one strain. The pangenome estimated size was 6552 (Chao lower bound) and open with an estimated alpha coefficient of 0.70 (Fig. 3b), and the genomic fluidity between pairs of genomes was estimated to be approximately 0.0679 (S.D.: 0.022, number of simulations 10,000), which showed not only a high degree of overlap between genomes regarding gene cluster carriage but also the openness to assimilating novel determinants that might play a role in niche adaptation.

A principal component analysis (PCA) based on the accessory genome showed agreement with the clustering resulting from the SNP-based analysis. Furthermore, PC1 revealed a clear separation between the Asian phylogroups (SP and AP) (Fig. 3c) and the majority of the European phylogroups (EP2, EP3, EP4, and EP5), and PC2 showed lineage separation between both major lineages of EP5, namely, Lpg and Leq (Fig. 3d).

## Recombination events driving adaptation

Despite the clonal structure of the population, both the population description and the pangenome analysis revealed different patterns of gene content, which allowed us to further stratify the population into lineages and sublineages and confirmed that the acquisition/loss of traits is mostly mediated by MGEs. The chromosomally integrated MGEs carrying AMR and virulence genes were identified via structural analysis based on multiple sequence alignments of our de novo Swiss hybrid assemblies and the completely assembled genomes retrieved from the collection ($n = 28$). In addition to the already known chromosomal positions where the main drivers of population evolution and adaptation are located, namely, the *tet*(M)-carrying transposon Tn*916* (a defining feature of the livestock-associated MRSA), the *mecA*-carrying SCC*mec* and the human IEC-carrying φSa3 prophages[17], our analysis identified several other MGEs (i.e., prophages and genomic

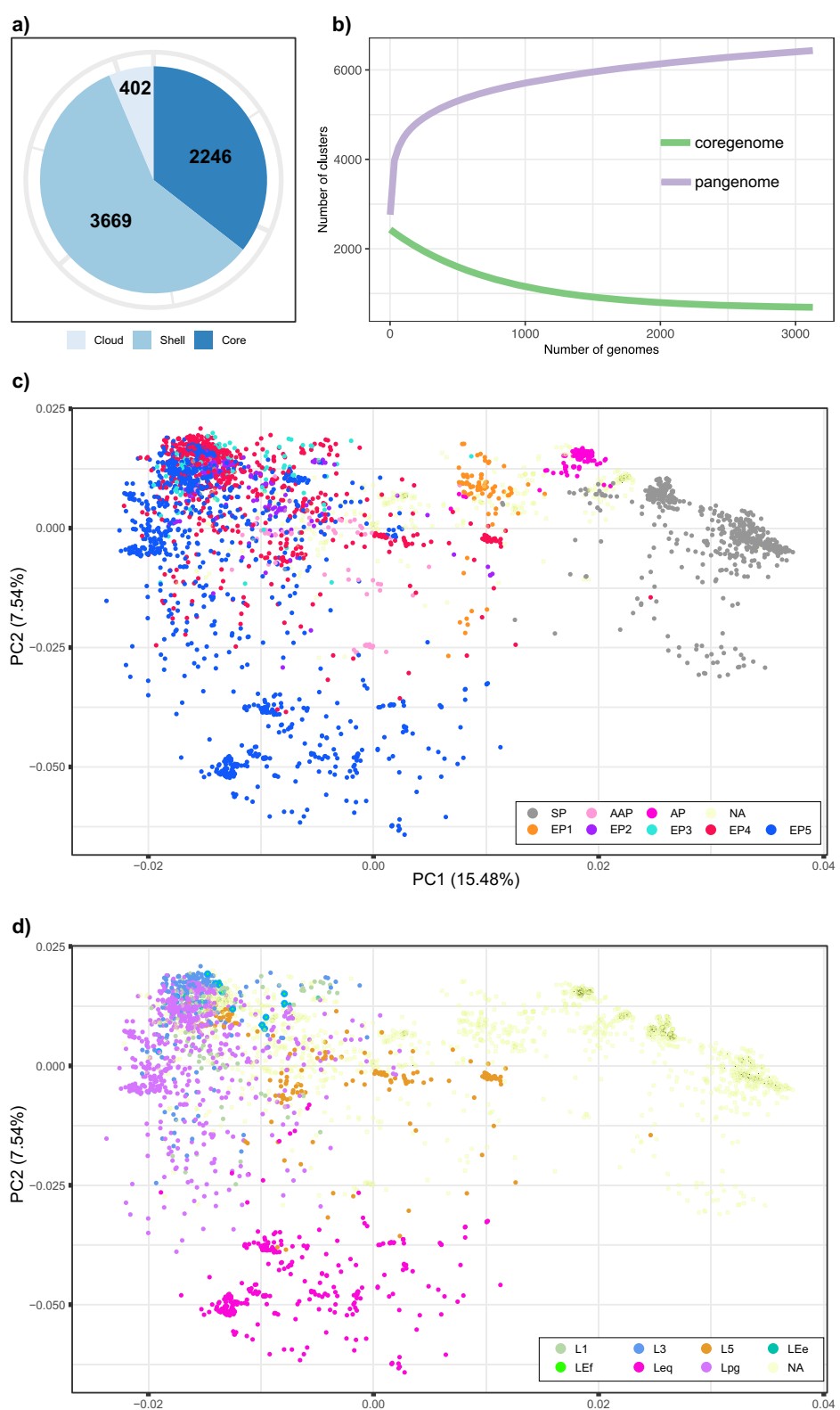

**Fig. 3 | Pangenome analysis of an international *Staphylococcus aureus* CC398 population. a** Bar pie showing the structure of the pangenome, core genes are those present in >95% of strains, shell genes are present in at least two strains, and cloud genes are singletons. **b** Rarefaction curves showing the number of unique clusters in the pangenome (violet curve) and the core genome (green curve) as a function of the number of genomes observed. **c**, **d** PCA of the pangenome matrix. Circles represent genomes in the two first principal components (directions) of the space, colors represent phylogroups in **c**, and major lineages in **d**. Percentages indicate how much of the total variation is observed within each principal component.

islands) and two additional recombination hotspots within the CC398 genome.

## Prophages

Identification of prophages of the (formerly known as) *Siphoviridae* family, which are the most common in *S. aureus*, was performed by in silico screening for phage integrase genes of the most common types, namely, Sa1int (representative prophage φ55), Sa2int (φ47), Sa3int (φ42E), Sa4int (φSauS-IPLA35), Sa5int (φ29), Sa6int (φ77), Sa7int (φ53), Sa8int (φSauS-IPLA88), Sa9int (vB_SauS_Mh1) and Sa12int (MR11), followed by structural comparisons of some of the groups identified. The most abundant prophages were those containing integrase types 3 ($n = 1099$), 2 ($n = 1050$), 6 ($n = 1001$), 9 ($n = 693$) and 1 ($n = 690$). Less prevalent were those that possessed integrase types 7 ($n = 64$), 5 ($n = 49$), 4 ($n = 28$), 12 ($n = 15$), and 8 ($n = 6$). In this collection of *S. aureus* CC398 strains, the majority (93,3%, $n = 2919$) carried at least one prophage, 1130 (36,1%) carried two prophages, 279 (8,9%) harbored three prophages, 19 (<1%) possessed four prophages, and 7 (<1%) carried five different prophages. Notably, 209 (6,7%) of the isolates had no prophages integrated into their chromosome (Fig. 4b). The Sa1int prophages were predominantly found (i.e., in more than 50% of the strains) in EP3, L3 (EP4) and Leq (EP5); Sa2int were mostly identified in EP1, EP2 and Lpg (EP5); Sa6int, in c1 and EP3; most of the lineages of EP4 (except L5) and Leq (EP5); and Sa3int prophages were the most common type among the human-clade strains (Fig. 4a).

The prophages varied in terms of structure and gene content, and the insertion sites in the CC398 chromosome were quite conserved among the prophage integrase types. Sa1int prophages were mostly integrated at the 3′ end of the iron-sulfur (Fe-S) assembly protein-coding gene *sufB*; Sa2int prophages, at the 3′ end of the *srrB* gene; Sa3int, in the bacterial sphingomyelinase gene *hlb*, disrupting its expression (thus called *hlb*-converting bacteriophages); Sa4int, at the 3′ end of the *ygdQ* gene; Sa5int, near the 5′ end of the *queE* gene; Sa6int, 141 bp upstream of the 3′ end of the lipase-encoding gene *geh*; and Sa7int, at the intergenic region between the 5′ end of the *isdB* gene and the 3′ end of the *rpmF* gene. Sa8int and Sa9int were commonly found downstream of the 3′ end of a tRNA-Ser coding gene, and Sa9int prophages were also found integrated at the 3′ terminus of the tmRNA-coding gene *ssrA*, which is a common insertion for phages in Gram-negative bacteria. Sa12int was integrated at the 3′ end of the *rclA* gene.

In addition to Sa3int, whose spread among *S. aureus* CC398 strains has been widely documented and associated with virulence and host adaptation[6], less common virulence-carrying prophages (i.e., non-Sa3int) were identified and further analyzed to determine the variability in structure and gene content. Sa1int phages were the fourth most abundant, and almost 70% of them ($n = 482/690$) carried the immune evasion factor gene *tarP* (Fig. 4a), which encodes a protein that glycosylates wall teichoic acid polymers, preventing them from becoming immunogenic and preventing host defense mechanisms from acting. The majority of *tarP*-Sa1int prophages ($n = 335/482$) were carried by EP5 strains, which suggests that after the acquisition, most of the lineages remained stable. However, in some sublineages, such as LEd, *tarP*-Sa1int was lost, and in LEg, a 110-bp deletion event occurred in the gene encoding the phage transcriptional activator RinA, which likely suppressed the formation of infectious phages[37]. The TarP determinant was additionally found in 193 Sa9int, 23 Sa3int, 3 Sa2int-like, 3 Sa5int, and 1 Sa12int prophage (Fig. 4a). A closer comparison of their structures revealed that *tarP* was the only common genetic feature among them, with the remaining structures differing for each integrase type.

In addition to the *tarP*-carrying and Sa3int strains harboring IEC determinants, most of the prophages identified had very diverse mosaic genomes, and the majority did not carry any virulence determinants. A few exceptions were a Sa7int variant carried by three related strains of EP4 isolated from pigs in Denmark that harbored the enterotoxin A and the staphylokinase genes *sea* and *sak*; a Cre recombinase associated with a pig strain in Taiwan that harbored the *sak* and *hlb* (phospholipase C) genes; and a *tarP*-containing Sa1int prophage in a pig strain from Denmark (EP4-L1) that also harbored the *sea* and *sak* genes. This prophage exhibited a hybrid structure with the module containing Sa1int and *tarP* and a fragment of approximately 25 kbp containing *sea* and *sak*, which shared 98.38% identity with the Sa3int prophage of a strain from Russia that harbored the *sea-sak-scn* virulence factors (Fig. 4g). To our knowledge, this is the first report of a Sa1int prophage carrying these important virulence factors.

## Genomic islands

The in silico screening of this CC398 population revealed that 1335 (42.7%) strains carried at least one SaGI and that 50 (1.6%) of them harbored two. Several different SaGIs, including some not previously described, were identified. These SaGIs included integrases of one of four families (Supp. Fig. 1) that target four known CC398 chromosomal sites. The most common integrases belonged to the tyrosine recombinase family and targeted a site downstream of the *guaA* gene at minute 9 (9′, nomenclature in minutes taken from ref. 38) within the larger genomic island *v*Saα, which carries genes of a staphylococcal superantigen-like (ssl) cluster (Fig. 5). The most common SaPIs, namely, SaPIpig1 and SaPIpig2/SaPIbov5, were integrated at this site. SaPIpig1 was present in 435 (13.9%) strains, and SaPIpig2/SaPIbov5 was present in 706 (22.5%) strains. Hybrid SaPIpig1/2-like elements containing the integration/regulation module of SaPIpig1 and the virulence module of SaPIpig2 (SaPIpig1-2) or vice versa (SaPIpig2-1) were identified in 10 and four strains, respectively, revealing the plasticity and modularity of these pathogenicity islands. Notably, the SaPIpig1 isolated from a pig in Germany was inserted at a different chromosomal location truncating the zinc metalloproteinase aureolysin-encoding gene *aur*, indicating that the tyrosine recombinase integrase family responsible for mobilization of SaPIpig-like islands can also integrate at different sites in the chromosome. An additional 13,923-bp novel resistance island (SaRI), which shares the backbone of SaPIpig1 and harbors the tellurite resistance-associated gene *tehA* (Fig. 5), was found in 47 strains (Reference strain APHA03, GenBank acc. no. ERR1437617) from different countries and hosts. The second most common integrase family of SaGIs targets the intergenic region between the *ktrB* and *groL* genes (44′). SaPIs integrated at this site ($n = 105$) carried a variety of toxin-encoding genes, such as the enterotoxin genes *sea*, *seb*, *sec*, or the toxic-shock syndrome gene *tsst*, many of which have not been previously described. The third family of integrases targets a chromosomal site at the 3′ end of the *metQ* gene (19′). Thirteen SaGIs were identified at this position, 6 of which were SaPIs that carried the enterotoxin genes *sek* and *seq* (Fig. 5). The last family of integrases was located after the 3′ end of the *rpsR* gene, which encodes the small ribosomal subunit protein bS18 (8′). Among the eleven elements inserted at this site, seven were SaPIs harboring virulence genes such as *tsst* or *scn*. Notably, the 3′ end of the tyrosine recombinases mobilizing these SaGIs consists of a variable region of 1 to 4 kbps (additional variable end region−AVR), which may contain additional antimicrobial resistance genes. Among them, a novel SaRI was identified that carried the trimethoprim resistance gene *dfrG* flanked by IS*1* (Fig. 5).

## Recombination hotspots

Two chromosomal regions of CC398 were identified as hotspots for the insertion of variable transposable elements (TEs) (Fig. 6). The first hotspot was located at the 3′ end of the *radC* gene, where at least six different transposable elements were identified within the livestock-associated clade (Fig. 6a). All these TEs had a core of three transposition-associated genes (*tnp*-like) and a variable accessory gene content. The transposon Tn*558*, which carries the florfenicol-chloramphenicol exporter gene *fexA*, was found predominantly in

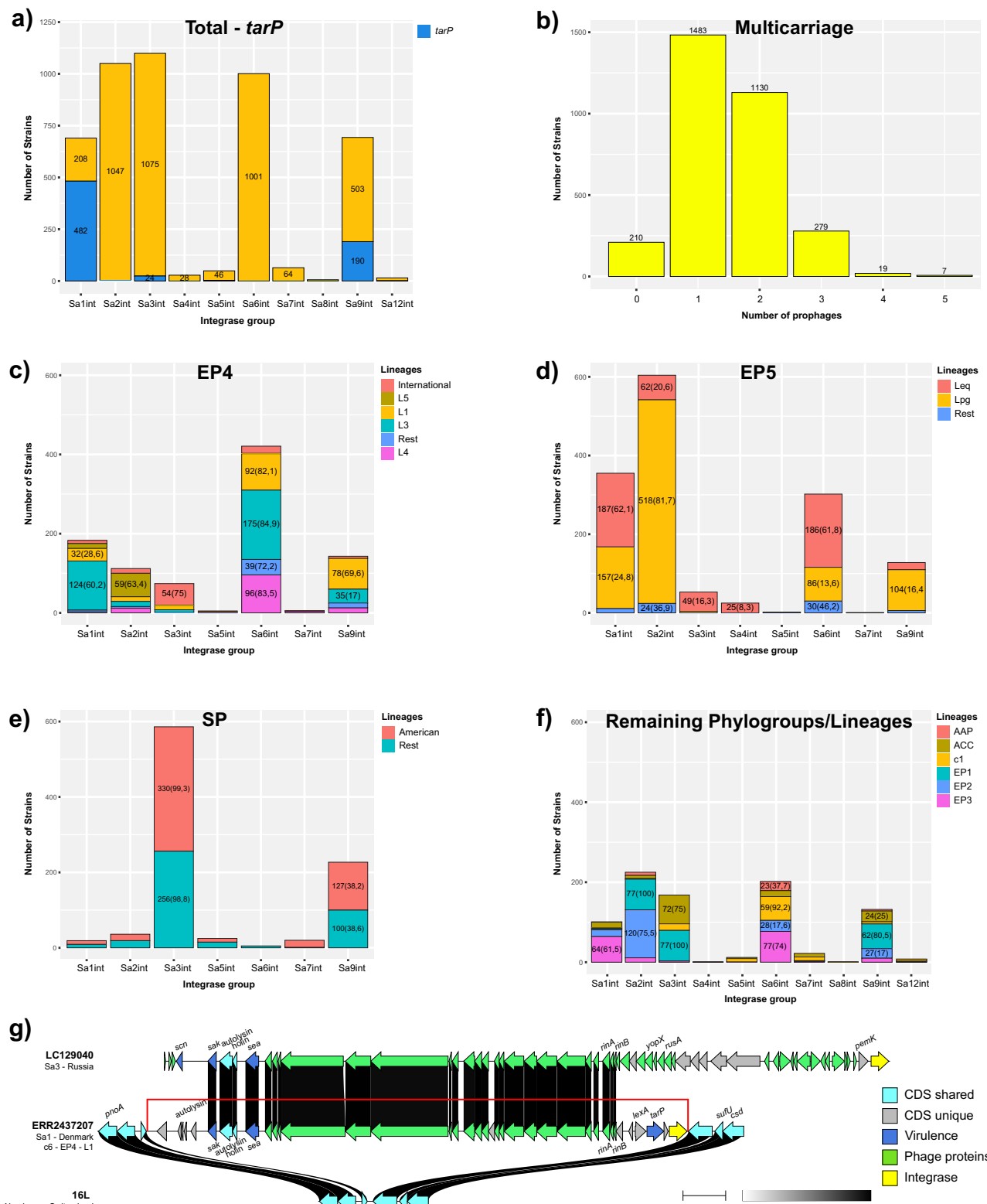

**Fig. 4 | Prophage carriage and distribution among *Staphylococcus aureus* CC398 population. a** Total carriage of different integrase families-associated prophages and a subtotal of those harboring the virulence factor *tarP*. **b** Multicarriage of prophages. **c**, **d**, **e**, and **f** Detailed, lineage subdivided carriage of prophages of major phylogroups and the rest of the population studied, namely EP4, EP5, SP, and the remaining strains, respectively. **g** Hybrid *tarP*-Sa1int prophage harboring *sea* and *sak* virulence factors.

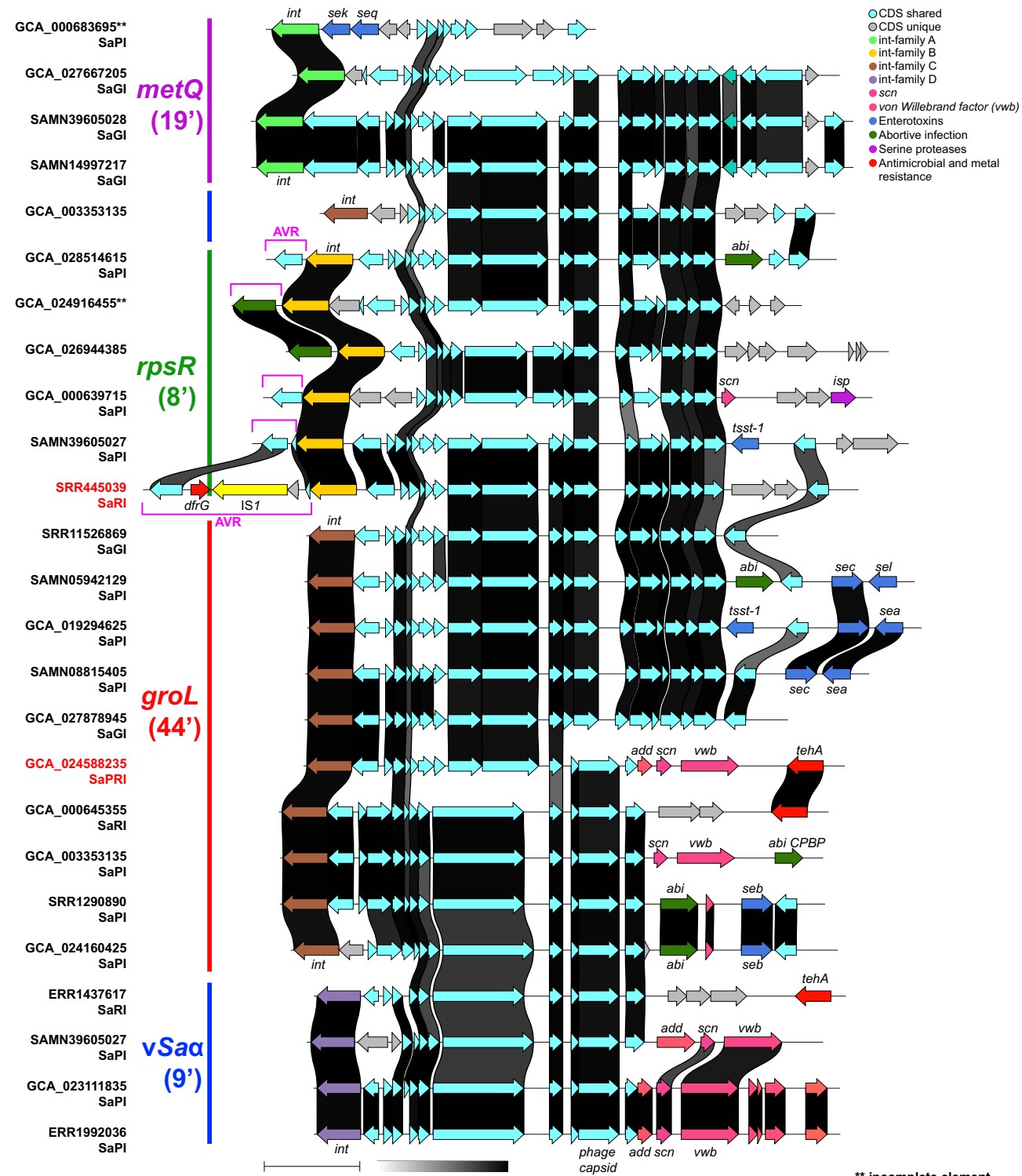

**Fig. 5 | *Staphylococcus aureus* Genomic Islands (SaGI) structural variability among the CC398 population.** Pairwise analysis of CC398 SaGIs, including a novel *dfrG*-containing SaRI and several not previously described enterotoxins harboring SaPIs. Image constructed using clinker v.0.0.28. CDS coding sequence, IS insertion sequence, *int* integrase, *scn* staphylococcal complement inhibitor, SaPI *S. aureus* Pathogenicity Island, SaRI *S. aureus* Resistance Island, SaPRI *S. aureus* Pathogenicity and Resistance Island, AVR additional variable end region.

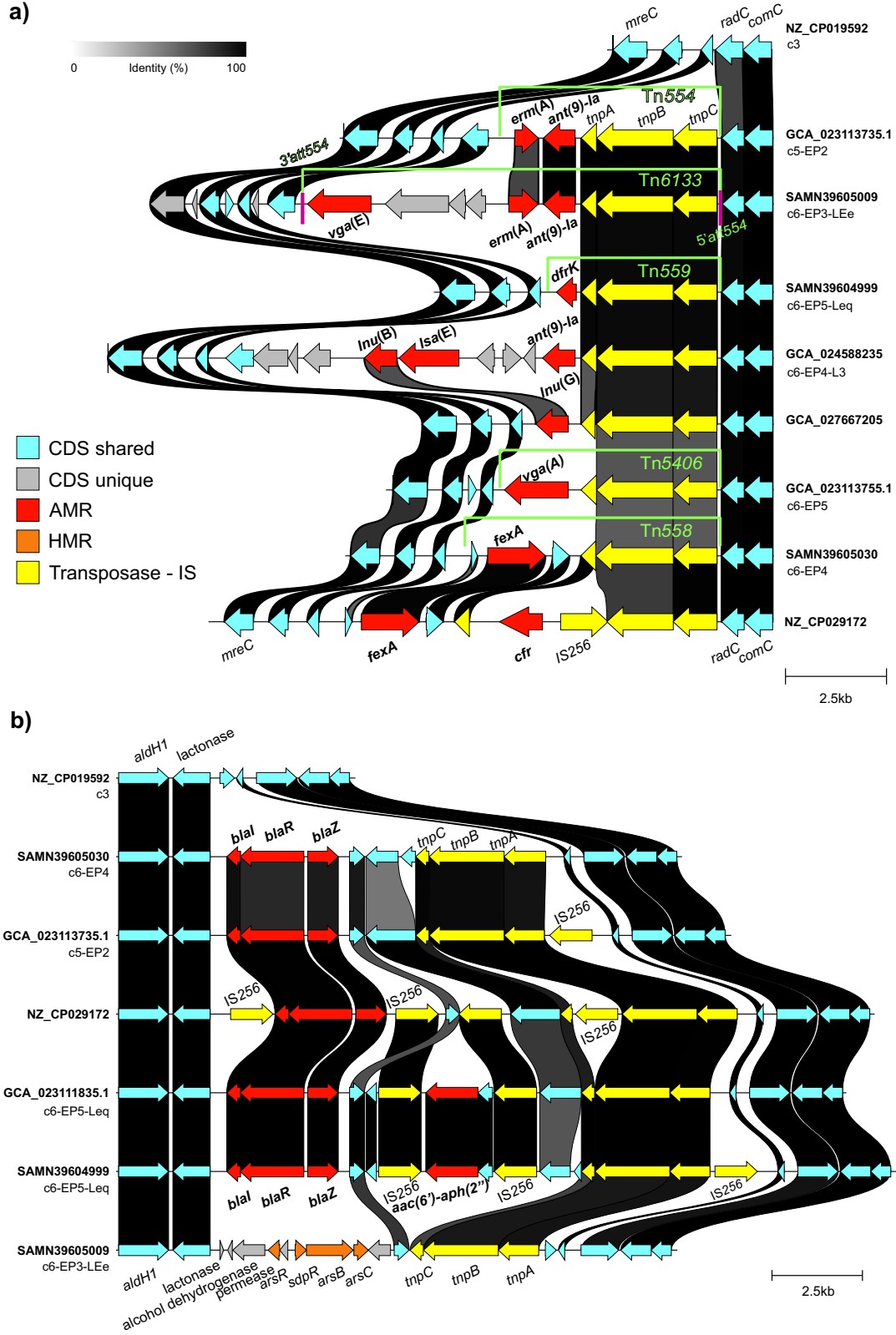

**Fig. 6 | Mobile genetic elements carrying AMR and virulence genes inserted at the two major recombination hotspots identified in CC398.** Pairwise comparison of representative strains from identified phylogroups and lineages to identify recombinant fragments involved in the resulting differences observed in AMR and virulence profiles. Transposons inserted at the hotspot downstream of *radC* are shown in **a** and those downstream of a lactonase gene in **b**. Image constructed using clinker v.0.0.28. CDS coding sequence, AMR antimicrobial resistance, HMR heavy metal resistance, IS insertion sequence, Tn transposon, *att* attachment site, c1–6 clades 1 to 6, EP1-5 European phylogroups 1 to 5, Leq equine lineage, LE European lineages, L3 Danish lineage 3.

strains of AAP, and in some lineages, it also carried the 23 S rRNA methyltransferase gene *cfr* (which provides resistance to chlor-amphenicol, florfenicol and linezolid) mobilized by IS*256* (Fig. 6a). The transposon Tn*554*, which harbors the MLS$_B$ resistance gene *erm*(A) and the spectinomycin 9-adenyltransferase gene *ant(9)-Ia*, was pre-dominantly present in strains from EP1. The Tn*554* variant Tn*6133*, which additionally contains the lincosamide, streptogramin A, and pleuromutilin (LS$_A$P) *vga*(E) genes, is characteristic of EP3 strains. The transposon Tn*5406*, which carries the LS$_A$P resistance gene *vga*(A), was found predominantly in strains from the L5 lineage of EP4 (Fig. 6a).

The second hotspot was located at the 5' end of the lactonase gene, where TEs of different lengths and with different gene contents were inserted to truncate the *yolD* gene (Fig. 6b). For instance, strains from EP2, EP4, and EP5 carried Tn*554*-like transposons consisting of a backbone formed by the gene cluster *blaI-blaR-blaZ* and transposase genes. The majority of strains from EP5-Leq (*n* = 271/301) carried the same transposon backbone but with an additional module carrying the aminoglycoside resistance gene *aac(6')-Ie−aph(2'')-Ia* flanked by IS*256* inserted upstream of *blaZ*, and 96% (*n* = 97/101) of the EP3 strains had, instead of the resistance genes, an 8898 bp fragment containing an arsenic resistance gene cluster (*arsR*, *arsB*, *arsC*). Importantly, these arsenic-resistant strains of EP3 also carried *blaZ*, as part of a novel SCC*mec* type Vc variant. This novel variant acquired an additional recombinant fragment (9836 bp) that carries the cluster *blaI-blaR-blaZ* and the trimethoprim resistance gene *dfrG* (Fig. 7).

## Discussion

We performed an in-depth molecular characterization of a worldwide collection of *S. aureus* CC398 strains. A timed phylogeny based on a global molecular clock was reconstructed using the genomes of five continental regions-spanning collection of MSSA and MRSA CC398 strains. The available metadata, such as the isolation date and location, permitted the consideration of geographic and temporal dimensions to enrich the phylogenetic description. This time-calibrated whole-genome-based phylogeny showed the presence of five phylogroups, EP1 to EP5, which spread across Europe, diverged decades ago from other MSSA lineages prevalent in Asia and North America, and further evolved over time. These phylogroups were defined based on the identification of a specific epidemiologically relevant fingerprint and included several related lineages and sub-lineages, some of which were identified and described in other European countries, such as the Danish and Italian lineages[39], as well as novel lineages (Lpg and Leq). For instance, Leq consisted of MRSA CC398-t011-IVa, which was predominantly isolated from horses and humans, and Lpg by MRSA CC398-t011-Vc, which was isolated from pigs, cattle, and humans. Each of the lineages also further evolved independently into sublineages according to their origin and type of host. Such geographic, host-related, and time-framed analyses allowed for the determination of relatedness among strains of diverse hosts and geographical origins. Under the assumption of a coalescent population evolving according to a molecular clock which was

consistent but lower than reported before for *S. aureus*, the clustering of strains from different geographical origins indicates international spread and the introduction of successful foreign CC398 clones into local populations. For instance, several strains isolated from pigs in Switzerland clustered within the EP4 lineages L1 and L3, which are mainly formed by MRSA CC398-t034-Vc isolated from pigs in Denmark and Hungary. The close relatedness of strains from Denmark, Hungary, and Switzerland indicates that a recent common ancestor was likely spread through animal trading. Indeed, a recent study from Hungary demonstrated the genetic relatedness between the MRSA CC398 strains from pigs from Denmark and those from farms in Hungary that imported pigs from Denmark[40]. Although Switzerland has an indigenous pig production system, a few hundred animals are imported each year for breeding purposes, which could also have contributed to the introduction of the L1 and L3 lineages into Swiss pig production. Otherwise, the majority of MRSA CC398 strains (>90%) isolated from pigs and cattle in Switzerland belonged to specific dis-tant new lineages of clade 6 (Lpg in EP5 and LEa in EP3) indicating several introductions of MRSA CC398 in these animal populations. The presence of these novel lineages may also suggest that MRSA CC398 of Lpg and LEa evolved independently from MSSA in Switzerland. How-ever, the absence of MSSA CC398 genomic data from pigs and cattle did not permit to verify this eventuality.

MRSA CC398 isolated from horses in Germany, Switzerland, Denmark, and the Netherlands belonged to the new specific lineage Leq within EP5. Crucially, MRSA of CC398-t011-IVa, which is the defining feature of Leq, has been detected and associated with sporadic outbreaks in equine settings all around Europe, including countries such as Austria[41], Hungary[42], Spain[43], the UK[44], France[45], and the Netherlands[46,47], which highlights a broader dissemination poten-tial. Obtaining WGS data from previously identified but not yet sequenced as well as newly detected MRSA CC398-t011-IVa would allow us to map them into this phylogeny, refine the population description, and identify specific evolution of the Leq lineage. Thus, identifying global and local adaptation. Such an adaptation was observed particularly in the sublineage LEg in Germany, which acquired the *eq*IEC and has not only been isolated from horses but also from cattle. Furthermore, further studies are necessary to identify specific genetic traits that may contribute to the successful coloniza-tion and infection of equines by Leq strains and to understand their potential to be successful among other host species including humans. Indeed, genetic relatedness between MRSA CC398 strains from ani-mals and those from humans was observed for each lineage and genetic evidence supported the idea of recent transmissions, a phe-nomenon mainly associated with occupational risk exposure, as already documented in several studies[39,48–50]. Notably, most human MRSA strains (carriage and clinical) in Switzerland as well as a con-siderable number in the Netherlands were CC398-t011-IVa and were closely related to strains from the horse-associated lineage (Leq). Considering that people working with horses are at risk of also being colonized and the fact that the exchange of clinic personnel from

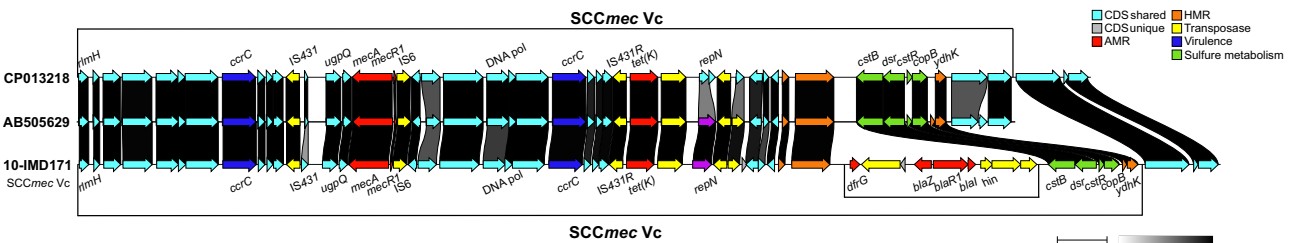

**Fig. 7 | Novel SCC*mec* Vc (5C2&5) variant which carries the resistance deter-minants *dfrG* and *blaZ* identified in lineage c6-EP3.** Pairwise nucleotide com-parison of the SCC*mec* element from a representative strain from c6-EP3 as well as the reference SCC*mec* V extracted from the GenBank. Graphical representation created with clinker v.0.0.28. CDS coding sequence, AMR antimicrobial resistance, HMR heavy metal resistance.

different countries through, for instance, residency and exchange programs between different universities and clinics are very common during academic formation, horse clinic personnel may contribute to the introduction of MRSA Leq into novel equine clinics and further dissemination to the community[51,52]. Livestock workers and veterinarians are also known to be at higher risk of being colonized and developing LA-MRSA infections[15], however, further metadata are necessary to determine whether humans who developed infections with horse-related MRSA strains were at occupational risk or had contact with people associated with horses or other animals. Importantly, longitudinal studies are required to better understand if the cases of MRSA CC398 human carriage are due to transient or long-term host adaptation-mediated colonization. Although the reconstructed time-framed phylogeny indicated that spillover events occurred often and recently due to the close relatedness among strains, additional metadata would confirm this phenomenon. The creation of specific MRSA CC398 international databases aiming to share WGS, as well as metadata in a harmonized way, would facilitate a quicker and more effective analysis of the dynamics of *S. aureus* at a local and global level.

The in silico screening for AMR, virulence, and MGE further revealed the genetic features characterizing the strains of each lineage. The horse-associated MRSA strains of lineage EP5-Leq (CC398-t011-IVa) exhibited similar antimicrobial and virulence patterns. Most EP5 strains contained either SaPIpig1 or SaPIpig2/SaPIbov5, which are two variants of SaPIbov4. These SaPIs harbor the genes *scn* and *vwb* which code for the *Staphylococcus* complement inhibitor and the von Willebrand factor-binding proteins, respectively[8,53]. While *scn* may contribute to improved colonization and adaptation in humans, the *vwb* proteins of SaPIpig1, SaPIpig2, and SaPIbov4 have been shown to coagulate the plasma of several animals, including pigs, sheep, and bovines, but not equine plasma[8,53]. The equine-related *scn-eq* variant mobilized by the φSaeq1 prophage, which also carries the equine-specific leukocidin-encoding gene *lukPQ*, was only carried by MRSA from an Leq German sublineage (LEg) circulating in horses and cattle that also harbored a *scn*-encoding SaPI. The absence of additional equine-adapted genes suggests that additional unidentified factors may be responsible for adaptation and pathogenicity in both horses and humans. Nevertheless, the presence of *scn* may explain the ability of MRSA CC398 of lineage Leq to easily colonize and be maintained in the personnel working with horses with the risk of causing opportunistic infections in humans. On the other hand, SaPIpig2 and SaPIbov4 were not present in MRSA CC398 from pigs and cattle in Switzerland, although they were originally identified in porcine and bovine MRSA in other countries[8,53–55], indicating the complex dynamics at play when following the evolution and adaptation of MRSA CC398 lineages and further highlighting local evolution a lineage.

In addition to factors that may have contributed to host adaptation, selection may also have been driven by long-term usage of antibiotics and the presence of antimicrobial resistance genes specific to the different lineages of MRSA CC398. For instance, the *aac(6')-le-aph(2'')-Ia* tandem gene was exclusively detected among MRSA strains of the horse lineage Leq and confers resistance to aminoglycosides such as gentamicin, which is commonly used in combination with β-lactams in equine medicine[56,57]. Antimicrobial resistance profiles are also very well conserved among the strains of the different lineages, further emphasizing clonal dissemination. Notably, both the β-lactam and aminoglycoside resistance genes were located on the same mobile genetic element in some MRSA strains of Leq. Similarly, the multidrug resistance transposon Tn*6133*, which contains the lincosamide-pleuromutilin resistance gene *vga*(E), the MLS$_B$ resistance gene *erm*(A), and the spectinomycin nucleotidyltransferase gene *ant(9)-Ia*[58], was only present in livestock MRSA from EP3. These antibiotics have been commonly used to treat infectious diseases in livestock, and their use may have contributed to the selection and maintenance of ST398-

t034/t011-Vc harboring Tn*6133* in food-producing animals. Notably, MRSA ST398 of EP3 also harbors a novel SCC*mec*Vc element, which is characterized by the integration of the β-lactamase gene *blaZ* and the trimethoprim-resistant dihydrofolate reductase gene *dfrG*, further providing a broader selective advantage for maintenance of MRSA in the animal population when undergoing antimicrobial treatment. Importantly, a recent study has shown that MGEs carrying resistance and virulence genes such as transposons and phages may contribute to bacterial survival in blood and host adaptation[59], indicating that AMR use might not be the only significant factor influencing the selection of MRSA CC398 and that MGE may have an impact in their ability to adapt to different host species.

This study provided a broad overview of the structure, phylodynamic trajectory, and chromosome-associated mobilome of the global population of CC398 *S. aureus* with a focus on the emergence and evolution of MRSA CC398. Considering the worldwide historical situation, lineages derived from both European phylogroups were identified in Switzerland. The global expansion of CC398 MRSA among livestock and horses during recent decades has been driven mainly by the expansion of the animal-adapted clade number 6 and specifically two major successful phylogroups (EP4 and EP5) that further diverged forming new lineages with specific AMR and adaptive traits. These results also confirmed the spread and adaptation of MRSA CC398 among livestock and the occurrence of spillover events between hosts and, crucially, humans, colonizing professionals at risk but also causing human infections. Nevertheless, a current limitation of this study is the restricted availability of MRSA CC398 WGS data from certain regions such as Africa, North America, or certain European and Asian countries. Data from European countries such as Denmark, the Netherlands, Germany, or Italy are abundant due to the fact that LA-MRSA CC398 has emerged and spread remarkably in their territories and that these countries have used a WGS-based approach to study the phenomenon. MRSA CC398 WGS data from other countries may so far not be available due to either the absence of MRSA CC398 in the animal and human population, lack of WGS-based MRSA surveillance programs, or absence of WGS-based analyses. Therefore, the reconstructed phylogeny obtained in this study is based on current available WGS data, resulting in some countries and regions being either overrepresented or underrepresented, thus highlighting the need for international collaborations to better understand the global expansion by refining this analysis and closely monitoring the evolution and spread of MRSA CC398. Importantly, this study reconfirmed previous findings and added novel data, providing an updated baseline for future epidemiological studies, which will permit us to determine at the local level whether the introduction of a strain into a clinical or farm setting is recent and what is the most likely origin of the strain. Additionally, such a WGS-based baseline could be used to further monitor the global expansion of some lineages in regions where MRSA CC398 may emerge or its presence has so far not been investigated. Therefore, continuous, and collaborative WGS-based long-term monitoring and surveillance of MRSA from a global One Health perspective are necessary to follow the evolutionary dynamics of MRSA CC398 and to identify successful and emerging lineages, as well as the evolutionary processes that gave rise to them, which may further spread and cause infections in both animals and humans.

## Methods

### Ethics statement
The samples from food-producing animals were taken at the slaughterhouse and the animal clinical samples were taken by veterinarians for diagnostic purposes therefore not requiring ethical approval or a permit for animal experimentation according to the current Swiss legislation (Federal Animal Protection Law, 455 (https://www.fedlex.admin.ch/eli/cc/2008/414/en).

## Origin of bacterial strains and whole-genome sequences

A total of 134 CC398 MRSA strains representing all isolated from animals, meat, and people working with animals in Switzerland between 2007 and 2020 were obtained from the Center for Zoonoses, Animal Bacterial Diseases and Antimicrobial Resistance (ZOBA) at the University of Bern and from previous research projects. The animal MRSA strains originated from samples taken at slaughterhouses from the nasal cavities of healthy cattle ($n = 21$) and pigs ($n = 23$); from chicken meat ($n = 10$) and pork meat taken at retail ($n = 3$); from infection sites of dogs ($n = 1$) and cats ($n = 1$)[60]; from infection sites as well as nasal cavities of horses ($n = 24$) (ref. 51 and Institute collection) and from the environment ($n = 2$). MRSA from healthy people working with animals originated from the nasal cavities of veterinary healthcare providers ($n = 23$) and farmers ($n = 6$)[27,48,51]. The clinical human strains ($n = 20$) isolated from patients in the German-speaking part of Switzerland were already sequenced, and the WGS reads were obtained from the SPSP. Genomic data from previous international MRSA CC398 studies as well as all novel available (to the best of our knowledge) WGS reads or assembled genomes of international *S. aureus* CC398 strains were downloaded from the Sequence Read Archive (SRA) or the National Center for Biotechnology Information (NCBI) genome database. After the identification and removal of duplicates, the collection consisted of sequences from 2994 *S. aureus* CC398 strains (2046 MRSA and 948 MSSA) from diverse countries spanning five continental regions. The aim was to obtain a collection of MRSA CC398 (over MSSA) covering all parts of the world with available data. However, in certain regions (e.g., America or Asia) CC398 is not the predominant MRSA lineage and most of the available data belonged to MSSA CC398. As already shown by previous studies, MRSA CC398 evolved from MSSA so to understand the evolutionary dynamics and the geographical spread of CC398 we included MSSA data from underrepresented MRSA regions to increase the scope of the analysis. The available metadata, including geographical origin, host, genetic characteristics, BioProjects, or GenBank/SRA accession numbers of the *S. aureus* CC398 strains used in this study, are listed in Supplementary Data 1.

## Whole-genome sequencing and assembly

The genome sequences of the MRSA CC398 Swiss strains were obtained from Nextera libraries prepared with the Illumina® DNA Prep (M) Tagmentation Kit (Illumina). Libraries were sequenced on an Illumina system to obtain $2 \times 150$ bp paired-end short reads at the Next Generation Sequencing Platform, Institute of Genetics, Vetsuisse Faculty, University of Bern. The remaining WGS reads were downloaded from the SRA (see above). Two isolates from each of the three prevalent Swiss lineages identified were additionally sequenced using Nanopore MinION long-read technology to obtain a complete assembled circular genome for use as a reference scaffold for comparative analysis. WGS reads that were not previously assembled were processed and analyzed using Trimommatic[61] for the initial steps of removing sequencing adaptors and quality-based filtering of the raw reads with a window size of 20 bp and a quality threshold of 30. Subsequent assembly was performed using the normal mode of the Unicycler v0.4.9 assembler[62] with default parameters. The six-lineage representative Swiss strains that were sequenced using both short- and long-read technologies were assembled using the hybrid assembly mode of Unicycler v0.4.9 with default parameters.

## Time-calibrated phylogeny

A total of 3128 genome sequences, including those of the 136 *S. aureus* CC398 strains from Switzerland and the 2992 international strains, were subjected to comparative genomic analysis, and a time-calibrated phylogeny was constructed. All the de novo assemblies were aligned against the MRSA ST398 type strain reference genome (GenBank accession number NC_017333.1), and Snippy version 4.4.5 (https://github.com/tseemann/snippy) was used to call the core genome single nucleotide

polymorphisms (cg-SNPs). Recombination events were detected and filtered out from the alignments using Gubbins version 2.4.1[63]. Before inferring the time-calibrated phylogeny, the existence of a temporal signal in the data was assessed and confirmed by using TempEst v1.5.3[64] and Clockor2[65] which supported the utilization of a global clock (Supp. Figs. 2 and 3). Subsequently, based on the recombination-free cgSNPs (19,899 sites) obtained previously, a timed phylogenetic tree was constructed using BEAST2[66]. For model selection, we used nested sampling (NS)[67] to choose between the strict and optimized relaxed molecular clock models. The XML files were created with BEAUti2 using the isolation date, a GTR site, a strict clock (based on the NS results), and two different coalescent population models, constant and exponential (both with a log-normal prior). To account for constant sites, we used beast2_constsites (https://github.com/andersgs/beast2_constsites) to add this information and obtain the final XML file. The MCMC was set to run for 300 million steps logging every 10,000 with a 10% burn-in. Each MCMC configuration was run twice to confirm convergence, and completeness was assessed by checking the Effective Sample Size of each parameter of interest to be above 200 with Tracer v1.7.2[68]. The results from both runs for each configuration were combined, and based on Bayes factor analysis, the model with a strict clock and a coalescent exponential population was chosen as the best model for fitting the data. A maximum clade credibility tree was constructed using TreeAnnotator v2.7.1 and edited with iTOL[69].

## *spa* and SCC*mec* typing

The assembled contigs were analyzed with the commercial software suite Ridom SeqSphere+ version 8.5.1[70] for *spa* typing. SCC*mec* typing was performed using the the staphophia-sccmec standalone version 1.0.0 from the Staphophia pipeline[71] for all samples and the SCCmecFinder 1.2 at the Center for Genomic Epidemiology, Technical University of Denmark (https://cge.food.dtu.dk/services/SCCmecFinder/)[72] for the Swiss MRSA and those that were not successfully typed by staphophia-sccmec.

## Antimicrobial resistance and virulence gene profiling

Antimicrobial resistance (AMR) and virulence determinants were screened in silico using the abricate tool against Resfinder[73] and the Virulence Factor Database (VFDB)[74], which assess presence based on nucleotide similarity. AMR findings were reconfirmed using Resistance Gene Identifier (RGI) software based on the CARD database[75]. For some resistance and virulence factor genes that are not recognized by the abovementioned software (e.g., SaPI-associated virulence factors), manual screening was performed using BLAST.

## Population structure

To establish a reference framework that would expand the current knowledge about CC398 population structure in Europe and consider the important role that recombination events have played in the evolution and adaptation of *S. aureus* populations, we defined a criterion that considered the genetic similarity and divergence time among strains as well as epidemiologically relevant information such as molecular characteristics (*spa* and SCC*mec* type), antimicrobial resistance (AMR) and virulence profile, geographical location, isolation host and available information from previous studies (i.e., lineage definitions). As a result, we established three nested layers to describe the population structure by means of clades, phylogroups, and lineages. Clades refer to the products of major divergence events that took place more than 45 years ago; phylogroups, are those that had specific genomic and epidemiological characteristics; and lineages, which denote smaller clades that diverged more recently (less than 45 years ago).

## Pangenome reconstruction

All CC398 genomes were annotated using Prokka with default parameters[76], and the graph-based clustering tool Panaroo[77] was

subsequently utilized to reconstruct the pangenome using the "strict" mode to remove putative contamination and erroneous annotations. After running the pipeline, we filtered out the sequences that were of unusual length or were flagged as pseudogenes from the resulting presence/absence matrix using the provided algorithm (panaroo-filter-pa), and the results with associated metadata were analyzed in R using the Pagoo framework to execute a principal component analysis (PCA), define the structure of the pangenome[78], and use the micropan package[79] to estimate pangenome size and openness according to Heap's law model and compute genomic fluidity.

### Identification of lysogenic prophages and genomic islands
To detect lysogenic prophages and SaGI, a manual approach termed "nested screening" was used and proceeded as follows. Known *S. aureus* phage and SaPI integrases available in the Nucleotide database of the NCBI were extracted including phage integrases Sa1int (representative prophage φ55; NC_007060, YP_240491), Sa2int (φ47; NC_007054, YP_240030), Sa3int (φ42E; NC_007052, YP_239890), Sa4int (φSauS-IPLA35; NC_011612, YP_002332364), Sa5int (φ29; NC_007061, YP_240566), Sa6int (φ77; NC_005356, NP_958623), Sa7int (φ53; NC_007049, YP_239679), Sa8int (φSauS-IPLA88; NC_011614, YP_002332477), Sa9int (vB_SauS_Mh1; OM439673, UKM36468.1) and Sa12int (MR11; NC_010147, YP_001604091.1) as well as SaPI integrases from SaPIpig1 (MW589252, QXJ78686.1), SaPIbov (AF217235, AAG29618.1), SaPIbov4 (HM211303, ADN95150.1), SaPIbov5 (HM228919, ADN53643.1), Ishikawa11 (AB716350, BAM66903.1), Hirosaki4 (AB716352, BAM66938.1), SaPIj11 (AB704541, BAM66859.1), IVM10 (AB716349, BAM66883.1), SaPIhhms2 (AB704540, BAM66842.1), and an initial in silico screening using BLAST (blastn/tblastn)[80] was performed for their presence using a threshold for positive identification of >90% coverage and >90% identity, as well as manual inspection of known insertion sites along the chromosome, as described previously by Novick et al.[81,82] (*Step 1*). Fragments containing an integrase with its flanking regions were extracted from positive strains in the collection and aligned with the same fragment (i.e., flanking regions) from a SaGI-negative strain to identify the whole structure of the elements and the exact site in the chromosome where they were inserted (*Step 2*). A region spanning 2-3 "representative" genes (*add frag*) was extracted from each element, and a second screening was performed. Strains that had a hit for an integrase/flanking region but not for the *add frag* region were manually inspected, allowing for the identification of novel structures (*Step 3*). Step 3 was iteratively performed for all the elements initially identified, and novel "representative" regions were identified for further screening.

### Structural analysis
Complete or semicomplete assemblies (i.e., those that possessed contigs with complete MGEs and flanking chromosomal regions) were subjected to pairwise and multiple comparisons via the progressive algorithm of Mauve[83] to describe in detail the structure of the identified prophages and SaGIs. For the final visualization of complete elements, we used Clinker v.0.0.28[84].

### Integrase gene relatedness
A multiple alignment of the nucleotide sequences of 22 SaGI integrases was performed using Clustal Omega 1.2.2[85], and a phylogenetic tree based on the alignment was reconstructed using RAxML 8.2.11[86] with the rapid hill-climbing algorithm and a GTR GAMMA I substitution model.

### Statistics and reproducibility
No statistical method was used to predetermine the sample size. No data were excluded from the analyses.

### Reporting summary
Further information on research design is available in the Nature Portfolio Reporting Summary linked to this article.

## Data availability
The genomic data of all the strains used in this study are available in the NCBI GenBank and NCBI Sequence Read Archive databases under accession numbers listed in Supplementary Data 1. Other relevant data supporting the findings presented are available within the paper and its Supplementary material.

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

## Acknowledgements

We are grateful to Alexandra Collaud (Institute of Veterinary Bacteriol-ogy, University of Bern, Switzerland) and Helena Seth-Smith (Institute of Medical Microbiology, University of Zurich, Switzerland) for their tech-nical support. We also thank Vinzenz Gerber and his team for providing horse clinic samples. This study was financed by the Swiss National Science Foundation (SNSF) grant no. 177504 "Development of a Swiss surveillance database for molecular epidemiology of multidrug-resistant pathogens" within the National Research Program NRP72 'Antimicrobial Resistance' to A.E. and V.P. and by internal funds (REF-660-50) of the Institute of Veterinary Bacteriology, University of Bern, Switzerland to V.P.

## Author contributions

The project was conceived by V.P. and J.E.F. The data analysis was performed by J.E.F. under the supervision of V.P. G.O. provided strains and data for LA-MRSA from food-producing animals. A.E. supervised and provided the WGS data of the clinical human strains. The manuscript was drafted and edited by J.E.F. and V.P. and revised by all the authors.

## Competing interests

The authors declare no competing interests.

## Additional information

Javier Eduardo Fernandez ⓘ¹, Adrian Egli², Gudrun Overesch ⓘ³ & Vincent Perreten ⓘ¹ ✉

¹Division of Molecular Bacterial Epidemiology and Infectious Diseases, Institute of Veterinary Bacteriology, Vetsuisse Faculty, University of Bern, Bern, Switzerland. ²Institute of Medical Microbiology, University of Zurich, Zurich, Switzerland. ³Center for Zoonoses, Animal Bacterial Diseases and Antimicrobial Resistance, Institute of Veterinary Bacteriology, Vetsuisse Faculty, University of Bern, Bern, Switzerland. ✉e-mail: vincent.perreten@unibe.ch

