## [Peer review file · Nature Communications]

REVIEWER COMMENTS

Reviewer #1 (Remarks to the Author):

The authors aimed to reconstruct a temporally calibrated phylogeny for the *S. aureus* clonal complex CC398 based on a collection of more than 3000 strains isolated over a period of 30 years. The genome data of 2994 *S. aureus* strains were obtained from SRA or the NCBI database. 134 isolates from Switzerland obtained between 2007 and 2020 from various animals, food and clinical human samples were also sequenced. In addition, metadata was included in the analyses as far as available.

Data were further subjected to in silico screens to search for resistance and virulence genes, identify the associated mobile genetic elements (phages, genomic islands, transposable elements) as well as their insertion sites and to compare their structure in order to determine associations between specific patterns and certain lineages or hosts.

The study is characterized by its broad strain collection and comprehensive analyses, which make it possible to draw a detailed picture of the global CC398 population, which may help to quickly identify newly emerging strains and prevent their spread.

The manuscript is well structured and written clearly. I am not a bioinformatician, but from my perspective, methods used are comprehensibly selected and the results generated are supported by a set of well-designed and clear illustrations.

Specific comments:

l. 51 Why did you put “and humans” in brackets?

l. 134 “...all the strains from EP1 as well as those from c1, c2, c3 and c4 harbored an IEC (chp135 sak-scN in EP1) and a PVL (except two strains)...”

I am confused by this sentence. I see from Fig. 2 that all isolates from c1, c2, c3 and c4 harbor an IEC, but not all of those harbor PVL. Please clarify this sentence.

l. 105/115/123/141/144/150

Please check numbers of isolates in c1 to c6. Adding these together results in more than 700 isolates.

l.144 (“ instead of “;”?

l. 157 c6 clade instead of “gene”?

l. 452 ff.

Do you think your data collection is suitable to draw conclusions on associations between human and animal strains?

l. 462-464

Please check sentence.

l. 476

Please check reference.

l. 481-485

Please check sentence.

Fig. 1

The color scheme of the time rings makes it difficult to recognize the branching of the clades. c1-c6 is not explained.

Fig. 2

Mutations in *gyrA* and *grlA* are given in detail; what 's about *rpoB*? What do the red/orange bars at *rpoB*, *hly*, *SaPIpig2* mean? Some more information about the color coding (presence/absence) would be helpful in the figure legend.

Fig. 4g

Legend is missing.

Fig. 6

Color code and legend includes "virulence", but there are no virulence genes depicted.

Reviewer #2 (Remarks to the Author):

The work conducted by Fernandez et al. demonstrates the importance of genomic surveillance of *Staphylococcus aureus* CC398, a lineage strongly associated with livestock but recently observed to infect humans as well. Within this paper, they employ time-calibrated phylogenetic analysis using a large global dataset (n=3,392) to determine the emergence of MRSA CC398 strains, which have been found among horses, pigs, and humans in Europe. However, it is already well known that the emergence of MRSA CC398 expansion is driven by prophages and other mobile genetic elements (PMID: 34828356). This paper reconfirms this but provides context as to when this might have occurred within Europe and gives a lot of detail of the structure of mobile genetic elements that drove this expansion.

Overall, the paper is very descriptive but requires refinements. The results section discusses all isolates analysed, whereas the discussion and abstract only focuses on the European strains. Perhaps some of the results can be summarised in a supplementary table (in an Excel file that has already been supplied) while maintaining focus on the European isolates and the most important

antimicrobial resistance (AMR), virulence, and mobile elements to make it more concise. Moreover, parts of the discussion are overly summarising the results rather than discussing the impact of their findings.

Regarding the actual analysis of the time-calibrated phylogeny on 3,128 samples, I wonder if the authors were able to obtain a good molecular clock reading that would have provided accurate results. They stated they used TempEst but did not report the results of a root-to-tip regression to determine if there is a temporal signal in the phylogenetic tree.

Other comments:

Line 476 has "[REF]". Remove and/or add missing reference.

Please state at line 630 the actual fragment used.

Reviewer #3 (Remarks to the Author):

This is a very thorough manuscript examining the evolution of CC398 strains in an international context. Overall the authors have anticipated many of my questions, but a few remain as described.

Methods: are these the totality of strains reported in this country during the time period? Also since the paper examines both MRSA and MSSA, are there any MSSA CC398 that were excluded from this dataset?

Similarly for the international dataset--assume this was the totality of strains included in these databases? Is it possible there were any duplicates between the two databases?

What are the strengths and limitations (especially the latter) of this work? The analyses are well-described but needs a bit more detail to put it into context, especially regarding missing data from countries that do not do this type of sampling or those that might be over-represented.

RESPONSE TO REVIEWERS

Reviewer #1 (Remarks to the Author):

The authors aimed to reconstruct a temporally calibrated phylogeny for the *S. aureus* clonal complex CC398 based on a collection of more than 3000 strains isolated over a period of 30 years. The genome data of 2994 *S. aureus* strains were obtained from SRA or the NCBI database. 134 isolates from Switzerland obtained between 2007 and 2020 from various animals, food and clinical human samples were also sequenced. In addition, metadata was included in the analyses as far as available.

Data were further subjected to in silico screens to search for resistance and virulence genes, identify the associated mobile genetic elements (phages, genomic islands, transposable elements) as well as their insertion sites and to compare their structure in order to determine associations between specific patterns and certain lineages or hosts.

The study is characterized by its broad strain collection and comprehensive analyses, which make it possible to draw a detailed picture of the global CC398 population, which may help to quickly identify newly emerging strains and prevent their spread.

The manuscript is well structured and written clearly. I am not a bioinformatician, but from my perspective, methods used are comprehensibly selected and the results generated are supported by a set of well-designed and clear illustrations.

We thank the reviewer for the time and constructive comments. Each comment is addressed below, and modifications reflected accordingly in the manuscript.

Specific comments:

I. 51 Why did you put “and humans” in brackets?

Response: Thanks for pointing it out, indeed, the brackets were not needed. We deleted the brackets in line 51: “*Animals and humans are...*”.

I. 134 “...all the strains from EP1 as well as those from c1, c2, c3 and c4 harbored an IEC (chp135 sak-scn in EP1) and a PVL (except two strains)...”

I am confused by this sentence. I see from Fig. 2 that all isolates from c1, c2, c3 and c4 harbor an IEC, but not all of those harbor PVL. Please clarify this sentence.

Response: Thanks for identifying this unclear sentence. The aim was to describe EP1. As you pointed out all of those (EP1, c1, c2, c3, and c4) harbored an IEC and the PVL was carried by **all** EP1 strains (after re-checking). We have reformulated the sentence focusing only on EP1 in lines 128-130 as follows: “*Notably, all the strains from EP1 harbored an IEC (chp-sak-scn), the PVL and lacked the tet(M)-carrying transposon Tn916, which are characteristic features of the so-called CC398 human adapted clade.*”

I. 105/115/123/141/144/150

Please check numbers of isolates in c1 to c6. Adding these together results in more than 7000 isolates.

Response: Thanks for bringing this up. We double checked the numbers and definitions. The numbers in the previous definition were correct: $c1 = 64$, $c2 = 591$, $c3 = 355$, $c4 = 2117$, $c5 = 2090$, and $c6 = 1846$ and the issue resulting from the addition of all of them came from the fact that $c5$ and $c6$ were part of the wider $c4$, so the number 2117 corresponding to $c4$ already included $c5$ and $c6$, we acknowledged that this definition might be confusing. To avoid this issue, we have refined the cluster definition focusing on the major and relevant clusters. Now it was added the specification of the jump to livestock and $c4$, $c5$ and $c6$ were redefined. Specifically, $c4$ was redefined and corresponds to the group of 27 strains geographically disperse (Fig. 1 and Fig. 2), $c5$ corresponds to EP2, and $c6$ to the major one which diverged into three branches, one with AAP and EP3, and the remaining ones with EP4 and EP5. It is worth nothing that following the reviewer's 2 recommendations to focus on relevant European related findings, there are small clusters to which no name was assigned, however upon expansion of this phylogeny by the addition of strains (crucially from different not well covered regions) this smaller clusters might become more relevant. The final numbers after the update are as follow: $c1 = 64$, $c2 = 591$, $c3 = 355$, $c4 = 27$, $c5 = 159$, and $c6 = 1844$.

I.144 (“ instead of “;”?)

Response: Thanks for this correction. However, due to the reformulation of some parts of the manuscript, this sentence is no longer in the revised manuscript.

I. 157 $c6$ clade instead of “gene”?

Response: Thanks for this correction, indeed it is clade 6. We corrected and adapted the sentence in line 145: *“The first branch of clade 6 further split forming two phylogroups, the AAP (American-Asian phylogroup) and EP3”*.

I. 452 ff.

Do you think your data collection is suitable to draw conclusions on associations between human and animal strains?

Response: Thanks for bringing this topic up since we believe is one of the main advantages of using WGS for epidemiological studies. From a phylogenetic point of view, we believe that the data is suitable to assess the association for certain lineages spreading in certain countries. The most plausible explanation for the small phylogenetic distance observed in certain lineages found in animals and humans such as Leq or Lpg, is a recent transmission event between species, however as we intend to highlight in the manuscript this serves as a baseline to be updated and refined. Moving forward it would be ideal to have wider access to patient/carrier metadata as well as animal trading activities between countries not only to assess the actual epidemiological link but also indirectly assess the degree in which CC398 is present among the community. This comment has been addressed in the manuscript as follows: *“Importantly, longitudinal studies are required to better understand if the cases of MRSA CC398 human carriage are due to transient or long-term host adaptation-mediated colonization. Although the reconstructed time-framed phylogeny indicated that spillover events occurred often and recently due to the close relatedness among strains, additional metadata would confirm this phenomenon.”*

I. 462-464

Please check sentence.

Response: The sentence has been checked and for better readability, we have reformulated the sentence in lines 471-473 as follows: "*Most EP5 strains contained either SaPIpig1 or SaPIpig2/SaPIbov5, which are two variants of SaPIbov4. These SaPIs harbor the genes scn and vwb which code for the Staphylococcus complement inhibitor and the von Willebrand factor-binding proteins, respectively.*"

I. 476

Please check reference.

Response: [REF] was deleted from the manuscript.

I. 481-485

Please check sentence.

Response: This specific sentence has been deleted during the review process

Fig. 1

The color scheme of the time rings makes it difficult to recognize the branching of the clades. c1-c6 is not explained.

Response: Thank you for pointing out these details. To improve the legibility of the figure, we have changed the color of the time rings (replace the blue), decreased their size, and slightly increased the size of the tree branches. We also explained c1-c6 and other key abbreviations in the figure legends.

Fig. 2

Mutations in *gyrA* and *grlA* are given in detail; what's about *rpoB*? What do the red/orange bars at *rpoB*, *hIb*, SaPIpig2 mean? Some more information about the color coding (presence/absence) would be helpful in the figure legend.

Response: Thank you for pointing this out. We have added to the legend of Figure 2 the information regarding mutations in the *rpoB* gene (mutation most frequent H481N and group the others in one category) and fixed the colors for *hIb* and SaPIpig2. Additionally, we have provided more information regarding presence/absence for each column of the figure and added the most relevant AMR and virulence information to Supplementary Figure 1.

Fig. 4g

Legend is missing.

Response: Thank you for highlighting this. We have added the missing legend: "*g) Hybrid tarP-Sa1int prophage harbouring sea and sak virulence factors.*"

Fig. 6

Color code and legend includes “virulence”, but there are no virulence genes depicted.

Response: Thank you for highlighting this. The figure indeed contains no virulence genes. We have deleted the color-coded virulence and have adapted the legend accordingly also adding the meaning of abbreviations used to improve legibility. “CDS: coding sequence; AMR: antimicrobial resistance; HMR: heavy metal resistance; IS: insertion sequence; Tn: transposon; c1-6: clades 1 to 6; EP1-5: European phylogroups 1 to 5; Leq: equine lineage; LE: European lineages; L3: Danish lineage 3.”

Reviewer #2 (Remarks to the Author):

The work conducted by Fernandez et al. demonstrates the importance of genomic surveillance of *Staphylococcus aureus* CC398, a lineage strongly associated with livestock but recently observed to infect humans as well. Within this paper, they employ time-calibrated phylogenetic analysis using a large global dataset (n=3,392) to determine the emergence of MRSA CC398 strains, which have been found among horses, pigs, and humans in Europe. However, it is already well known that the emergence of MRSA CC398 expansion is driven by prophages and other mobile genetic elements (PMID: 34828356). This paper reconfirms this but provides context as to when this might have occurred within Europe and gives a lot of detail of the structure of mobile genetic elements that drove this expansion.

Overall, the paper is very descriptive but requires refinements. The results section discusses all isolates analysed, whereas the discussion and abstract only focuses on the European strains. Perhaps some of the results can be summarised in a supplementary table (in an Excel file that has already been supplied) while maintaining focus on the European isolates and the most important antimicrobial resistance (AMR), virulence, and mobile elements to make it more concise. Moreover, parts of the discussion are overly summarising the results rather than discussing the impact of their findings.

Response: Thank you very much for your time and constructive feedback. We have addressed all your comments and modified the manuscript accordingly. We have also simplified the results section focusing mostly on the major clades and phylogroups. We have also added the results of the AMR and MGEs screening to Supp. Table 1 as suggested. We have also refined the discussion keeping only some summarizing part of the results to introduce the corresponding discussion. We have also added a section regarding data limitations of the study as well as the output provided, which enriches current knowledge and provides a baseline for a future expansion of the analysis as more data becomes available.

Regarding the actual analysis of the time-calibrated phylogeny on 3,128 samples, I wonder if the authors were able to obtain a good molecular clock reading that would have provided accurate results. They stated they used TempEst but did not report the results of a root-to-tip regression to determine if there is a temporal signal in the phylogenetic tree.

Response: Thanks for raising this question. Indeed, in addition to TempEst we have used

additional software such as Clockor2 (PMID: 38366939) to perform an initial exploratory analysis, remove duplicates and execute a root-to tip regression to check for the presence of a temporal signal in the data prior to proceed to reconstruct the time-calibrated phylogeny. We have created a new Supplementary Figure (Supp. Fig. 2) with a graphical depiction of the root-to-tip analysis showing the presence of a temporal signal with a global clock indicated by a slope (R^2) of 0.025, a Bayesian Information Criterion (BIC) of -41250, and predicting the intercept at 1956.515 which approximated the one obtained a posteriori (Supp. Fig. 2 - a). Additionally, using the same software we performed an exploratory search for putative local clocks using as settings 3 local clocks and a minimum of 500 isolates per group. The results shown in Supp. Fig. 2 – b, should be interpreted carefully since are estimative and tend to have overfitting (as it was well explained by the developers of the tool) however, they are indicators that demonstrate that for this dataset a global clock fits best the data.

The estimated molecular clock was found 95% higher posterior density (HPD) interval: $3.20-3.48 \times 10^{-7}$] which is consistent with but lower than that estimated in previous studies for *S. aureus* which indicates high conservation of the core genome of CC398.

We have also adapted the manuscript accordingly to reflect this data as follows in Methods: *“Before inferring the time-calibrated phylogeny, the existence of a temporal signal in the data was assessed and confirmed by using TempEst v1.5.3⁶¹ and Clockor2⁶² which supported the utilization of a global clock (Supp. Fig. 2)”*, in Results: *“Based on our analysis, the molecular clock of the CC398 lineage was estimated to be 3.34×10^{-7} substitutions per site per year [95% higher posterior density (HPD) interval: $3.20-3.48 \times 10^{-7}$] which is consistent with but lower than that estimated in previous studies for *S. aureus*^{31, 32, 33} and specifically CC398³⁴.”*, in Discussion: *“Under the assumption of a coalescent population evolving according to a molecular clock which was consistent but lower than reported before for *S. aureus*, the clustering of strains from different geographical origins indicates international spread and the introduction of successful foreign CC398 clones into local populations”*. and added the Supplementary Figure 2.

Other comments:

Line 476 has "[REF]". Remove and/or add missing reference.

Response: Thanks for pointing it out, [REF] was deleted from the manuscript.

Please state at line 630 the actual fragment used.

Response: We have added to the material and methods section the list of phages and SaPIs (and their integrases) that were used as reference as well as their accession numbers in the following format (phage/SaPI, integrase).

Reviewer #3 (Remarks to the Author):

This is a very thorough manuscript examining the evolution of CC398 strains in an international context. Overall the authors have anticipated many of my questions, but a few remain as described.

Thanks a lot for your time and constructive comments. We have responded to your questions below and modified the manuscript accordingly.

Methods: are these the totality of strains reported in this country during the time period? Also since the paper examines both MRSA and MSSA, are there any MSSA CC398 that were excluded from this dataset?

Response: Thank you for asking this precision. The Swiss MRSA CC398 isolated from animals, vets and farmers represent all isolated from these sources during the time period, however the human clinical strains were isolated from patients in the German speaking part of Switzerland. Regarding MSSA there were none purposely excluded. The aim was to gather a dataset from public databases with a predominance of MRSA CC398 (over MSSA) and to cover all parts of the world with available data (to the best of our knowledge). However, in certain regions (e.g., America or Asia) CC398 is not the predominant MRSA lineage and most of the available data belonged to MSSA CC398. As already shown by previous studies, MRSA CC398 evolved from MSSA so to understand the evolutionary dynamics and the geographical spread we understood that this data though from MSSA would provide data signal from these specific regions that would increase the relevance and scope of the analysis.

This information has been included into the Methods section of the manuscript as follows:

“A total of 134 CC398 MRSA strains representing all isolated from animals, meat and people working with animals in Switzerland between 2007 and 2020 were obtained from...”

“The clinical human strains (n=20) isolated from patients in the German speaking part of Switzerland were already sequenced...”

“Genomic data from previous international MRSA CC398 studies as well as all novel available (to the best of our knowledge) WGS reads or assembled genomes of international S. aureus CC398 strains were downloaded from the Sequence Read Archive (SRA) or the National Center for Biotechnology Information (NCBI) genome database. After identification and removal of duplicates, the collection consisted of sequences from 2994 S. aureus CC398 strains (2046 MRSA and 948 MSSA) from diverse countries spanning five continental regions. The aim was to obtain a collection of MRSA CC398 (over MSSA) covering all parts of the world with available data. However, in certain regions (e.g., America or Asia) CC398 is not the predominant MRSA lineage and most of the available data belonged to MSSA CC398. As already shown by previous studies, MRSA CC398 evolved from MSSA so to understand the evolutionary dynamics and the geographical spread of CC398 we included MSSA data from underrepresented MRSA regions to increase the scope of the analysis. ...”

Similarly for the international dataset--assume this was the totality of strains included in these databases? Is it possible there were any duplicates between the two databases?

Response: Thanks for expanding on this topic. Indeed, we selected all CC398 from these databases. Specifically, in the GenBank we thoroughly searched for all BioProjects containing CC398 and additionally we inspected extensively the literature for publications studying MRSA CC398 and include the samples referred to (either assembly or SRA). We acknowledge the fact that we might have missed some samples from the GenBank, now available but not at the time of executing the analysis or due to unintentionally missing them during the search process. In the preliminary exploratory analysis duplicates were removed.

This information has been included into the Methods section of the manuscript as follows:

“After identification and removal of duplicates, the collection consisted of sequences from 2994 S. aureus CC398 strains (2046 MRSA and 948 MSSA) from diverse countries spanning five continental regions ...”

What are the strengths and limitations (especially the latter) of this work? The analyses are well-described but needs a bit more detail to put it into context, especially regarding missing data from countries that do not do this type of sampling or those that might be over-represented.

Response: This is a very interesting aspect of the study. Among the strengths of it is the high number of samples included in the analysis which allowed to reconfirm and expand previous findings regarding the population structure and evolution of CC398. Crucially, the criteria defined to describe the population structure in clades, phylogroups and lineages considered factors that are temporally and epidemiologically relevant. For instance, besides the estimation of divergence time, also considering the presence/absence of AMR and virulence determinants as well as the MGEs involved in their mobilization added a valuable layer of evolutionary information to our understanding of the CC398 global population. On the other hand, a limiting factor for the study was the lack of equally distributed available WSG data. Certain European countries such as Denmark or the Netherlands were over-represented and at the same time data from other regions such as Africa or South America was scarce and clearly does not represent totally that local population. However, far from being a fixed or biased image, this analysis is an actual snapshot of the global population and provides an updated baseline for future efforts. The estimative nature of the analysis allows for the constant improvement and refinement of our understanding. The emergence of clusters, phylogroups, lineages and sub lineages were well supported by the genetic evidence, however this snapshot should be updated and refined periodically to better reflect the evolutionary trends of the population, involving the emergence and/or expansion of certain lineages due to host adaptation or favorable fitness factors as well as the decline of others. This information has been included into the Discussion section of the manuscript as follows: *“These results also confirmed the spread and adaptation of MRSA CC398 among livestock and the occurrence of spillover events between hosts and, crucially, humans, colonizing professionals at risk but also causing human infections. Nevertheless, a current limitation of this study is the restricted availability of MRSA CC398 WGS data from certain regions such as Africa, North America, or certain European and Asian countries. Data from European countries such as Denmark, the Netherlands, Germany, or Italy are abundant due to the fact that LA-MRSA CC398 has emerged and spread remarkably in their territories and that these countries have used a WGS-based approach to study the phenomenon. MRSA CC398 WGS data from other countries may so far not be available due to either absence of MRSA CC398 in the animal and human population, lack of WGS based MRSA surveillance programs, or absence of WGS-based analyses. Therefore, the reconstructed phylogeny obtained in this study is based on current available WGS data, resulting in some countries and regions being either overrepresented or underrepresented, thus highlighting the need for international collaborations to better understand the global expansion by refining this analysis and closely monitoring the evolution and spread of MRSA CC398. Importantly, this study reconfirmed previous findings and added novel data, providing an updated baseline for future epidemiological studies, which will permit to determine at the local level whether the introduction of a strain into a clinical or farm setting is recent and what is the most likely*

origin of the strain. Additionally, such WGS-based baseline could be used to further monitor the global expansion of some lineages in regions where MRSA CC398 may emerge or its presence have so far not been investigated. Therefore, continuous, and collaborative WGS-based long-term monitoring and surveillance of MRSA from a global One Health perspective are necessary to follow the evolutionary dynamics of MRSA CC398 and to identify successful and emerging lineages, as well as the evolutionary processes that gave rise to them, which may further spread and cause infections in both animals and humans”

We highly appreciate the feedback from all reviewers which helped us to remarkably improve the quality of the manuscript.

REVIEWERS' COMMENTS

Reviewer #2 (Remarks to the Author):

Satisfied with the changes by the author